# Non-Euclidean Mixture Model for Social Network Embedding

**Roshni G. Iyer**
Computer Science Department
University of California, Los Angeles
Los Angeles, California, USA
`roshnigiyer@cs.ucla.edu`

**Yewen Wang**
Computer Science Department
University of California, Los Angeles
Los Angeles, California, USA
`wyw10804@gmail.com`

**Wei Wang**
Computer Science Department
University of California, Los Angeles
Los Angeles, California, USA
`weiwang@cs.ucla.edu`

**Yizhou Sun**
Computer Science Department
University of California, Los Angeles
Los Angeles, California, USA
`yzsun@cs.ucla.edu`

## Abstract

It is largely agreed that social network links are formed due to either homophily or social influence. Inspired by this, we aim at understanding the generation of links via providing a novel embedding-based graph formation model. Different from existing graph representation learning, where link generation probabilities are defined as a simple function of the corresponding node embeddings, we model the link generation as a mixture model of the two factors. In addition, we model the homophily factor in spherical space and the influence factor in hyperbolic space to accommodate the fact that (1) homophily results in cycles and (2) influence results in hierarchies in networks. We also design a special projection to align these two spaces. We call this model Non-Euclidean Mixture Model, i.e., **NMM**. We further integrate **NMM** with our non-Euclidean graph variational autoencoder (VAE) framework, **NMM-GNN**. **NMM-GNN** learns embeddings through a unified framework which uses non-Euclidean GNN encoders, non-Euclidean Gaussian priors, a non-Euclidean decoder, and a novel space unification loss component to unify distinct non-Euclidean geometric spaces. Experiments on public datasets show **NMM-GNN** significantly outperforms state-of-the-art baselines on social network generation and classification tasks, demonstrating its ability to better *explain* how the social network is formed.

## 1 Introduction

Social networks are omnipresent because they are used for modeling interactions among users on social platforms. Social network analysis plays a key role in several applications, including detecting underlying communities among users [1], classifying people into meaningful social classes [2], and predicting user connectivity [3]. Most existing embedding models are designed based on the ***homophily*** aspect of social networks [4, 5]. They utilize the intuition that associated nodes in a social network imply feature similarity, and an edge is usually generated between similar nodes. Prior works have used shallow embedding models to represent homophily, like matrix factorization and random-walk (Section 2), which are parameter intensive and do not employ message passing. As an improvement, graph neural network (GNN) models (Section 2) have been proposed to more effectively capture homophily by representing a node through its local neighborhood context.

38th Conference on Neural Information Processing Systems (NeurIPS 2024).

However, research of **RaRE** [6] and work of [7] show homophily is insufficient, and ***social influence*** is also critical in forming connections. This is due to popular nodes having direct influence in forming links [8]. For example, in Twitter network, users tend to follow celebrities *in addition to* users who share similar interests [9]. Though **RaRE** jointly models both factors, it has limitations in modeling capabilities. Specifically, it is parameter intensive as each node embedding is fully parameterized through a Bayesian framework. Further, **RaRE** assumes graphs are transductive, limiting its performance in the practical inductive setting where new links not seen during training must be predicted. Moreover, nearly all works embedding social networks utilize a single zero-curvature Euclidean space, when in reality, network factors may create different topologies. Specifically, edges generated by homophily tend to form cycles [10], while edges generated by social influence tend to form tree structures [11, 12]. From Riemannian geometry, people have found that networks with cycles are best represented by spherical space embeddings [13], while tree structured networks are best represented by hyperbolic space embeddings [14]. Thus, an end-to-end model to bridge social network embeddings of distinct non-Euclidean geometric spaces is a promising direction.

Our motivation is two-fold: (1) We aim to *understand* how the social network is generated e.g., which factors affect node connectivity and what topological patterns emerge in the network as a result. (2) Using our learning from (1), we aim to design a more realistic deep learning model to *explain* how the network is generated (inferring new connections). We summarize our contributions as follows:

- We propose Graph-based Non-Euclidean Mixture Model (**NMM**) to explain social network generation. **NMM** represents nodes via joint influence by homophily (modeled in spherical space) and social influence (modeled in hyperbolic space), while seamlessly unifying embeddings via our space unification loss.

- To our knowledge, we are also the first to couple **NMM** with a graph-based VAE learning framework, **NMM-GNN**. Specifically, we introduce a novel non-Euclidean VAE framework where node embeddings are learned with a powerful encoder of GNNs using spherical and hyperbolic spaces, non-Euclidean Gaussian priors, and unified non-Euclidean optimization.

- Extensive experiments on several real-world datasets on large-scale social networks, Wikipedia networks, and attributed graphs demonstrate effectiveness of **NMM-GNN** in social network generation and classification, which outperforms state-of-the-art (SOTA) network embedding models.

## 2 Preliminary and Related Work

We provide an overview of social network embedding models and discuss advancements in non-Euclidean graph learning.

### 2.1 Social Network Embedding

Several works that embed social networks merely model homophily [15], capturing node-node similarity, without also considering a node's social influence or node popularity e.g., a celebrity. Homophily-based models include shallow embedding models and GNN embedding models. Most shallow embedding models are either based on matrix factorization [16] or random-walk [17]. Though GNN models [18] effectively learn on large networks and in inductive settings, they still fail to model the social influence factor in the network. Further, even the model that captures both homophily and social influence, **RaRE** [6], has limitations in that its fully-parameterized node embeddings require large parameter size to be learned, and it models all nodes in Euclidean space, an approach not effective in capturing different topologies (e.g., cycles and hierarchy) in the social network.

### 2.2 Non-Euclidean Geometry for Graphs

Non-Euclidean geometric spaces, commonly used to model surfaces in mathematics and physics, are curved geometries that include spherical spaces with positive curvature, and hyperbolic spaces with negative curvature [19]. Works have recently found Euclidean space modeling to be insufficient for non-Euclidean graph-structured data [20]. Namely, spherical spaces have been shown to effectively embed graphs with cyclic structure due to their positive curvature [21, 13], while hyperbolic spaces have been shown to effectively embed graphs with hierarchical structure due to their negative curvature and exponential or "tree-like" growth of the space [22, 23]. HGCN [24] and [25] are critical works exploring GNNs in non-Euclidean spaces. Both these hyperbolic GNN methods have shown

significant improvement on benchmark datasets by preserving hierarchical structure of graphs. In knowledge graphs, [26] has achieved notable performance by modeling graphs in non-Euclidean spaces of various curvatures, using both hyperbolic and spherical space.

# 3 Methodology

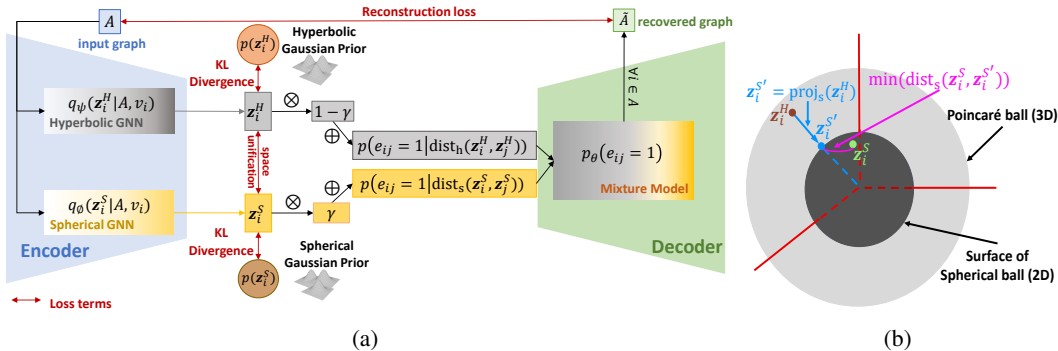

Figure 1: Model Architecture. (a) Architecture overview of **NMM-GNN**, a non-Euclidean Mixture Model with non-Euclidean VAE framework. (b) Illustration of space unification loss of **NMM-GNN** . $z_i^H$ from the hyperbolic space is projected to its corresponding point in the spherical space, $z_i^{S'}$, such that its geodesic distance is ensured to be close to $v_i$'s existing spherical space representation, $z_i^S$.

A social network $G = (V, A)$ consists of a set of vertices $V = \{v_i\}_{i=1}^N$, and associated adjacency matrix $e_{ij} \in A$, where $e_{ij}$ is an edge from $v_i$ to $v_j$. We aim to design a model to jointly learn both node homophily and social influence representation, denoted $z_i^S$ and $z_i^H$ respectively, that can best explain the social network in terms of link reconstruction. In this section, we describe our architecture. First, we introduce a novel non-Euclidean mixture model, called **NMM**, to model the probability of a new link. Second, we add a non-Euclidean GNN encoder to enrich **NMM**, called **NMM-GNN**, which is connected to the variational autoencoder framework. Source code is in the Appendix. For details on future directions, the reader is also referred to the Limitations section in the Appendix.

## 3.1 Overview

To model both factors of homophily and social influence that affect graph connectivity, we model each node $v_i$ with homophily regulated representation, $z_i^S$, and social influence regulated representation, $z_i^H$ [1]. A unified framework is designed such that $z_i^S$ and $z_i^H$ influence each other bi-directionally and are seamlessly merged through **NMM**. **NMM** utilizes non-Euclidean geometric spaces to better represent homophily and social influence components which produce curved structures like cycles and trees. To improve **NMM**, we enrich its encoder by GNNs, $q_\psi(z_i^H|G)$ and $q_\phi(z_i^S|G)$. To ensure generated embeddings are in spherical and hyperbolic spaces respectively, non-Euclidean GNNs are adopted. The enhanced model is **NMM-GNN**, which has a clear connection to the VAE framework.

Figure 1(a) illustrates the architecture overview of **NMM-GNN**. Specifically, the encoder component maps nodes into homophily based embedding, $z^S$, in spherical space (for homophily generated cycles) and social influence based embedding, $z^H$, in hyperbolic space (for social influence generated trees), which follow non-Euclidean prior distributions. The embeddings are passed into our mixture model decoder, which models the probability of a link as a mixture of a homophily based distribution component and a social influence based distribution component. The objective is to maximize the likelihood to observe the links, or equivalently to minimize the link reconstruction loss. In addition, the two geometric spaces are ensured to be aligned together via a space unification regularization term, to make sure the two embeddings of the same node are corresponding to each other.

## 3.2 Modeling via Non-Euclidean Mixture Model (NMM)

Geometrically, the space of **NMM** is a spherical surface inside a unit Poincaré ball. Each node $v_i$ corresponds to (1) a point in the Poincaré ball, $z_i^H$, and (2) a point on the spherical surface,

---

[1]The geometric space community conventionally uses $S$ to represent spherical space and $H$ to represent hyperbolic space.

$z_i^S$, where both spaces are embedded inside a Euclidean space. These two points are aligned by enforcing projection of $z_i^H$ onto the spherical surface to be close to $z_i^S$, as projecting a star onto earth's atmosphere, shown in Fig. 1(b). We note that the space alignment does not mean embeddings are enforced to be the same in two spaces[2]. Rather, we require the projection of $z_i^H$ is close to $i$'s embedding in spherical space $z_i^S$. In this case, the two distances of spherical and hyperbolic spaces are also different from each other (norm space difference vs. spherical geodesic distance). Without the space alignment, $z_i^H$ has too much degree of freedom, which can move freely as long its norm is kept the same. Lastly, the probability of generating a link is a mixture of the probability of generating the link in each non-Euclidean space.

### 3.2.1 Modeling for homophily

***Representation of homophily regulated nodes.*** We embed homophily based representation of node $v_i$, or $z_i^S$, on the surface of the spherical ball in spherical space, $\mathbb{S}^d$. Formally, $\mathbb{S}^d = \{z_i^S \in \mathbb{R}^d \big| \|z_i^S\| = w^S\}$ is the $d$-dimensional $w^S$-norm ball, where $\|\cdot\|$ is the Euclidean norm and $w^S \in [0, 1)$ is a constant to ensure the spherical surface is inside the unit Poincaré ball. To better capture homophily distribution, $p(z_i^S)$, we use spherical Gaussian distribution $G_S(\cdot)$ as spherical prior, described in Table 1. $\beta$ is the axis or *direction* of the lobe controlling where the lobe is located on the sphere, and points towards the center of the lobe; $\lambda$ is the *sharpness* of the lobe such that as this value increases, the lobe will become narrower in width; and $a$ is the amplitude or *intensity* of the lobe, corresponding to the height of the lobe at its peak.

***Link prediction using homophily based distribution.*** The probability of link $e_{ij} = 1$ between nodes $v_i$ and $v_j$, also described in Table 1, is determined by the geodesic distance between $z_i^S$ and $z_j^S$. $\text{dist}_s(z_i^S, z_j^S) = \arccos(\langle z_i^S, z_j^S \rangle)$ is the geodesic distance between $z_i^S$ and $z_j^S$; $J > 0$ and $B \geq 0$ are model parameters; and $\langle \cdot, \cdot \rangle$ denotes the inner product. The probabilistic model is designed so nodes greatly dissimilar (exhibiting low homophily), or $\text{dist}_s(z_i^S, z_j^S) \to +\infty$, have low probability of a link being generated, or $p_{\text{hom}}(e_{ij} = 1) \to 0$. In the case of nodes exhibiting high homophily, they either (1) may be connected due to highly similar characteristics, or (2) may not be connected simply because the individuals do not know each other. Our distribution models both scenarios. Note when $\text{dist}_s(z_i^S, z_j^S) \to 0$, $p_{\text{hom}}(e_{ij} = 1) \to \frac{1}{1 + e^B}$, which can be interpreted as a factor to control sparsity of the network.

Table 1: Homophily regulated nodes: Distribution Prior and Link Generation Probability

| (a) Spherical Distribution Prior | (b) Link Generation Probability |
|---|---|
| $p(z_i^S) = G_S(z_i^S; \beta, \lambda, a)$ | $p_{\text{hom}}(e_{ij} = 1) = p(e_{ij} = 1 \vert \text{dist}_s(z_i^S, z_j^S))$ |
| $= a e^{\lambda(\beta \cdot z_i^S - 1)}$ | $= \dfrac{1}{1 + e^{J \times \text{dist}_s(z_i^S, z_j^S) + B}}$ |

### 3.2.2 Modeling for social influence

***Representation of social influence regulated nodes.*** We embed social influence based representations of node $v_i$, or $z_i^H$, on the Poincaré ball from hyperbolic space, $\mathbb{H}^{d+1}$, to better capture resulting hierarchical structures. Social influence regulated nodes are represented as points, $z_i^H$, belonging inside the Poincaré (open) ball in $\mathbb{H}^{d+1}$. Formally, $\mathbb{H}^{d+1} = \{z_i^H \in \mathbb{R}^{d+1} \big| \|z_i^H\| = w_{z_i}^H\}$, $w_{z_i}^H \in [0, 1)$, is $(d+1)$-dimensional $w_{z_i}^H$-norm ball, where $\|\cdot\|$ is Euclidean norm. We assume center of the Poincaré ball is aligned with center of the sphere, and is one dimension larger than the sphere to ensure the spherical surface is inside the Poincaré ball. To better capture a social influence regulated distribution, $p(z_i^H)$, we use Hyperbolic Gaussian distribution $G_H(\cdot)$ as non-Euclidean Gaussian prior, described in Table 2. $\overline{z_i^H}$ is the origin of $(r, \omega)$ for radius $r$ and angle $\omega$ in polar coordinates. $\overline{z_i^H}$ is the center of mass and $\zeta > 0$ is the dispersion parameter, where the dispersion dependent normalization constant $Z(\zeta)$ accounts for the underlying non-Euclidean geometry. $Z(\zeta)$ is partitioned into angular $\omega$ and radial $r$ components. $\Gamma(\cdot)$ is Euler's gamma function, and $\text{dist}_H(\cdot)$ is the hyperbolic distance between two hyperbolic space node embeddings, $x$ and $y$, where $\|\cdot\|$ denotes Euclidean norm.

---

[2]Note that it is impossible to equate these two embeddings directly as they are in different geometric spaces.

***Link prediction using social influence based distribution.*** We model existence of an edge, $e_{ij} = 1$ between nodes $v_i$ and $v_j$, also described in Table 2, as a function of norm space difference, $\text{dist}_\text{h}(\cdot)$, between two hyperbolic space node embeddings, $z_i^H$ and $z_j^H$. We utilize norm space difference as opposed to hyperbolic distance, since nodes of similar social influence status may have large distance in the Poincaré (open) ball due to nodes possibly being placed towards the ball's boundary. This is because, at the boundary of the ball, nodes become infinitely distanced apart. Thus, to allow for numerical stability and to capture social influence difference (in which the higher the social influence of nodes indicated by large in-degree and smaller out-degree, the closer they are embedded towards the center of the ball), we utilize norm space as indicator. Consistent with the notation of [6], a node with higher social influence is associated with a smaller norm value. $C$ and $D$ are learned model parameters and norm space of a vector is $\text{norm}(\boldsymbol{x}) = \|\boldsymbol{x}\|$. $\text{dist}_\text{h}(z_i^H, z_j^H) = |\text{norm}(z_i^H) - \text{norm}(z_j^H)|$ is the norm difference between $z_i^H$ and $z_j^H$; $C > 0$ and $D \geq 0$ are model parameters; and $\text{norm}(\cdot)$ denotes the L1 normalization function. The probabilistic model is designed such that nodes largely different in popularity (exhibiting high social influence), or $\text{dist}_\text{h}(z_i^H, z_j^H) \to +\infty$, have high probability of a link being generated, or $p_\text{rank}(e_{ij} = 1) \to 1$. In the case both nodes exhibit low social influence (such as having similar social rank), they either (1) may be connected due to highly similar characteristics, or (2) may not be connected simply because individuals do not know each other. Our distribution models both scenarios. When $\text{dist}_\text{h}(z_i^H, z_j^H) \to 0$, $p_\text{rank}(e_{ij} = 1) \to \frac{e^D}{1+e^D}$, which can be interpreted as another factor to control sparsity of the network.

Table 2: Social influence regulated nodes: Distribution Prior and Link Generation Probability

(a) Hyperbolic Distribution Prior

$$p(\boldsymbol{z}_i^H) = G_H(\boldsymbol{z}_i^H; \overline{\boldsymbol{z}_i^H}, \zeta) \tag{1}$$

$$= \frac{1}{Z(\zeta)} e^{-\frac{\text{dist}_H(\boldsymbol{z}_i^H, \overline{\boldsymbol{z}_i^H})^2}{2\zeta^2}}$$

$$Z(\zeta) = Z_\omega(\zeta) Z_r(\zeta) \tag{2}$$

$$Z_\omega(\zeta) = \text{Vol}(\mathbb{H}^d) \tag{3}$$

$$= \frac{\pi^{\frac{d}{2}}}{\Gamma(\frac{d}{2} + 1)}$$

$$Z_r(\zeta) = \int_0^{+\infty} e^{-\frac{r^2}{2\zeta^2}} \sinh^d(r)\, dr \tag{4}$$

$$= \frac{1}{2^d} \sum_{k=0}^{d} \binom{d}{k} (-1)^k \sqrt{\frac{\pi}{2}} \zeta e^{\frac{(2k-d)^2 \zeta^2}{2}} \text{erfc}\left(\frac{(2k-d)\zeta}{\sqrt{2}}\right)$$

$$\text{dist}_H(\boldsymbol{x}, \boldsymbol{y}) = \text{arccosh}\left(1 + \frac{2\|\boldsymbol{x} - \boldsymbol{y}\|^2}{(1 - \|\boldsymbol{x}\|^2)(1 - \|\boldsymbol{y}\|^2)}\right) \tag{5}$$

(b) Link Generation Probability

$$p_\text{rank}(e_{ij} = 1)$$
$$= p(e_{ij} = 1 | \text{dist}_\text{h}(\boldsymbol{z}_i^H, \boldsymbol{z}_j^H))$$
$$= \frac{e^{C \times \text{dist}_\text{h}(\boldsymbol{z}_i^H, \boldsymbol{z}_j^H) + D}}{1 + e^{C \times \text{dist}_\text{h}(\boldsymbol{z}_i^H, \boldsymbol{z}_j^H) + D}}$$

### 3.2.3 Non-Euclidean Mixture Model

***Link Prediction Using Mixed Space Distribution.*** Since both homophily and social influence affect the connectivity structure of social networks, we model existence of a new link between nodes, $p_\theta(e_{ij} = 1)$, as a weighted combination distribution of these factors. Specifically, our non-Euclidean mixture model is a weighted combination of homophily based distribution, $p_\text{hom}(e_{ij} = 1)$, and social influence based distribution, $p_\text{rank}(e_{ij} = 1)$, with learned weight $\gamma$:

$$p_\theta(e_{ij} = 1) = p_\theta(e_{ij} = 1 | \boldsymbol{z}_i^S, \boldsymbol{z}_j^S, \boldsymbol{z}_i^H, \boldsymbol{z}_j^H)$$
$$= \gamma \cdot p_\text{hom}(e_{ij} = 1) + (1 - \gamma) \cdot p_\text{rank}(e_{ij} = 1) \tag{6}$$

where $p_\text{hom}(e_{ij} = 1)$ is modeled in positively curved spherical space since homophily based links may form cycles due to similarity connections between node clusters. $p_\text{rank}(e_{ij} = 1)$ is modeled in negatively curved hyperbolic space since social influence based links may form tree-like structures due to popularity-based social hierarchy between node clusters. We would like to highlight that the link between nodes $i$ and $j$ is a mixture model because each link is a weighted combination of influence from both spherical and hyperbolic spaces (not one or the other) as evidenced in Equation 6. As shown in Figure 1b, the same node has two representations – one in the spherical space and one in the hyperbolic space, and because they represent the same underlying node, they need to be aligned. Therefore, these two network factors do not contradict each other, but rather *work together* to explain how links are formed between users.

### 3.3 Modeling via Non-Euclidean VAE on Graphs

We enrich the encoder of **NMM** to generate better embeddings. To do so, we explore GNN methods which have been shown to be more effective than shallow embedding methods (see Experiments and Ablation Studies). We refer to this enriched **NMM** model as **NMM-GNN**. **NMM-GNN** uses non-Euclidean VAE as its learning framework. The framework integrates a mixture of different non-Euclidean geometric spaces e.g., hyperbolic and spherical spaces, for learning of encoder, decoder, and node prior distributions, and ensures geometric spaces are unified during training.

**Encoder model.** The encoder learns two corresponding embedding representations per node $v_i$ to produce spherical embedding $z_i^S$ and hyperbolic embedding $z_i^H$. For homophily regulated nodes in spherical space, $\mathbb{S}^d$, any spherical space GNN (SGNN) can be applied, and for social influence regulated nodes in hyperbolic space, $\mathbb{H}^{d+1}$, any hyperbolic space GNN (HGNN) can be applied. The general framework for SGNN and HGNN are shown in Tables 3 and 4. $z_i^{S^{(l)}} \in \mathbb{R}^{d^{(l)}}$ and $z_i^{H^{(l)}} \in \mathbb{R}^{d+1^{(l)}}$ are spherical and hyperbolic feature representations of node $v_i$ at layer $l$, with dimensionality $d$ and $(d+1)$ respectively. $f$ is a message-specific neural network function of incoming messages to $v_i$ from neighborhood context $N_i$, and activation function $\sigma(\cdot)$, typically $\mathrm{ReLU}(\cdot)$ for all layers but the last one being $\mathrm{softmax}(\cdot)$.

Table 3: General Framework and Example Model for Spherical Graph Neural Network.

| (a) General Framework: Spherical GNN | (b) Example: Spherical GCN |
|---|---|
| $\begin{aligned} z_i^{S^{(l+1)}} &= q_{\mathrm{SGNN}}(z_i^S|A, v_i) \\ &= q_\phi(z_i^S|A, v_i) \\ &= \sigma\big(\sum_{j \in N_i} f(z_i^{S^{(l)}}, z_j^{S^{(l)}})\big) \end{aligned}$ | $\begin{aligned} z_i^{S^{(l+1)}} &= q_\phi(z_i^S|A, v_i) \\ &= \exp_0^S\Big(\sigma\big(W_l^T\big(\sum_{j \in N_i \cup \{i\}} \frac{e_{j,i}}{\sqrt{m_j m_i}} \log_0^S(z_j^{S^{(l)}})\big)\big)\Big) \end{aligned}$ |

Table 4: General Framework and Example Model for Hyperbolic Graph Neural Network.

| (a) General Framework: Hyperbolic GNN | (b) Example: Hyperbolic GCN |
|---|---|
| $\begin{aligned} z_i^{H^{(l+1)}} &= q_{\mathrm{HGNN}}(z_i^H|A, v_i) \\ &= q_\psi(z_i^H|A, v_i) \\ &= \sigma\big(\sum_{j \in N_i} f(z_i^{H^{(l)}}, z_j^{H^{(l)}})\big) \end{aligned}$ | $\begin{aligned} z_i^{H^{(l+1)}} &= q_\psi(z_i^H|A, v_i) \\ &= \exp_0^H\Big(\sigma\big(W_l^T\big(\sum_{j \in N_i \cup \{i\}} \frac{e_{j,i}}{\sqrt{m_j m_i}} \log_0^H(z_j^{H^{(l)}})\big)\big)\Big) \end{aligned}$ |

***Example Non-Euclidean Encoder models.*** We illustrate Non-Euclidean Graph Convolutional Neural Network (Non-Euclidean GCN) as an example GNN. Tables 3 and 4 describe the model architectures for both Spherical GCN and Hyperbolic GCN respectively. Node features are initialized by sampling each embedding dimension from a distribution uniformly at random for all nodes where $z_i^{S^{(0)}} \in \mathbb{R}^{d^{(l)}} \leftarrow \mathrm{Unif}([0, 1))^d$ and $z_i^{H^{(0)}} \in \mathbb{R}^{d+1^{(l)}} \leftarrow \mathrm{Unif}([0, 1))^{d+1}$ respectively. The retraction operator, $\mathcal{R}(\cdot)$, involves mapping between spaces. For non-Euclidean spaces, retraction is performed between non-Euclidean space and approximate tangent Euclidean space using logarithmic and exponential map functions. Specifically, $\log_0^H(z_i^H) = \tanh^{-1}(\mathrm{i} \cdot \|z_i^H\|) \frac{z_i^H}{\mathrm{i} \cdot \|z_i^H\|}$ is a logarithmic map at center $\mathbf{0}$ from hyperbolic space to Euclidean tangent space, and $\exp_0^H(z_i^H) = \tanh(\mathrm{i} \cdot \|z_i^H\|) \frac{z_i^H}{\mathrm{i} \cdot \|z_i^H\|}$ is an exponential map at center $\mathbf{0}$ from Euclidean tangent space to hyperbolic space. $\log_0^S(z_i^S) = \tanh^{-1}(\|z_i^S\|) \frac{z_i^S}{\|z_i^S\|}$ is a logarithmic map at center $\mathbf{0}$ from spherical space to Euclidean tangent space and $\exp_0^S(z_i^S) = \tanh(\|z_i^S\|) \frac{z_i^S}{\|z_i^S\|}$ is an exponential map at center $\mathbf{0}$ from Euclidean tangent space to spherical space. where $z_i^{S^{(l)}}, z_i^{H^{(l)}}$ are embeddings of node $v_i$ at layer $l \in [0, L]$, $L = 2$; $W_l$ is a layer-specific learnable weight matrix; $N_i$ is the set of nodes in the neighborhood context of $v_i$; $e_{j,i}$ is the edge-weight between nodes $v_j \rightarrow v_i$, with default edge weight being 1.0 if an edge exists. $m_i, m_j$ are entries of the degree matrix, with $m_i = 1 + \sum_{j \in N_i} e_{j,i}$.

**Decoder model.** NMM can be considered as a probabilistic decoder for link generation, which maps embeddings of two nodes into the probablity to generate a link between them.

**Joint loss function.** The training loss involves components of reconstruction loss (to ensure the generated graph is consistent with the original graph), KL divergence loss (to ensure predicted embeddings $z_i^S$ and $z_i^H$ closely match their non-Euclidean Gaussian distributions), and space unification loss (to ensure $z_i^S$ and $z_i^H$ map to the same node $v_i$).

**Reconstruction Loss.** Table 5 shows reconstruction loss, which minimizes the upper bound on the negative log-likelihood. $\lambda_A$ is a hyperparameter; $A' = XAX^T$, given $X \in {0, 1}^{k \times d}$ where $X_{a,i} = 1$ only if node $a \in \tilde{G}$ is assigned to $i \in G$ and $X_{a,i} = 0$ otherwise, where $\tilde{G}$ is the predicted graph.

Table 5: Description for Reconstruction Loss.

$$p(A'|z^S, z^H) = \frac{1}{k(k-1)} \sum_{a \neq b} A'_{a,b} \log \tilde{A}_{a,b} + (1 - A'_{a,b})\log(1 - \tilde{A}_{a,b}) \quad (7)$$

$$-\log p_\theta(G|z_i^S, z_j^S, z_i^H, z_j^H) = -\lambda_A \log p(A'|z_i^S, z_j^S, z_i^H, z_j^H) \quad (8)$$

$$L_{\text{recon}}^S(\phi, \theta; G) = \mathbb{E}_{q_\phi(z_i^S|G)}[-\log p_\theta(G|z_i^S, z_j^S, z_i^H, z_j^H)] \quad (9)$$

$$L_{\text{recon}}^H(\psi, \theta; G) = \mathbb{E}_{q_\psi(z_i^H|G)}[-\log p_\theta(G|z_i^S, z_j^S, z_i^H, z_j^H)] \quad (10)$$

**KL Divergence Loss.** The KL divergence loss is formed by minimizing the equations described in Table 6. Minimizing the KL divergence loss ensures that the homophily regulated nodes and social influence regulated nodes closely align to their underlying non-Euclidean distribution priors. As described in Section 3, these distributions are designed to appropriately capture the distinct topologies that emerge as a result of the respective social network factors.

**Space Unification Loss.** The space unification loss is formed by minimizing the equations described in Table 6. Minimizing the space unification loss ensures that the hyperbolic space representation of node $v_i$ in spherical space, $\text{proj}_S(z_i^H)$, is close to the corresponding learned representation of node $v_i$ in the spherical space, $z_i^S$. Note that using a normalized hyperbolic disk is not a substitute for the projection operator from the social influence hyperbolic space onto the homophily spherical space. The projection operation *solely* projects out-of-sphere nodes onto the $w^S$ norm space, or the norm at the surface of the spherical ball. A normalization operator would instead change the embedding values of all nodes. More importantly, the space unification loss component ensures minimal spherical geodesic distance between $z_i^H$'s representation on the spherical ball and $z_i^S$, illustrated in Figure 1(b).

Table 6: Description of KL Divergence Loss and Space Unification Loss.

| (a) KL Divergence Loss | (b) Space Unification Loss |
|---|---|
| $L_{\text{KL}}^S(\phi; G) = \text{KL}[q_\phi(z_i^S|G)\|p(z_i^S)] \quad (11)$ 
 $L_{\text{KL}}^H(\psi; G) = \text{KL}[q_\psi(z_i^H|G)\|p(z_i^H)] \quad (12)$ | $\text{proj}_S(\boldsymbol{x}) = \begin{cases} w^S \cdot \frac{\boldsymbol{x}}{\|\boldsymbol{x}\|} & \text{if } \|\boldsymbol{x}\| \neq w^S \\ \boldsymbol{x} & \text{otherwise} \end{cases} \quad (13)$ 
 $L_{\text{unify}}(G) = \text{dist}_s(\text{proj}_S(z_i^H), z_i^S) \quad (14)$ |

**Total Loss.** The overall loss function for homophily regulated and social influence regulated nodes respectively is then a summation of the above loss components given by:

$$L^S = L_{\text{recon}}^S(\phi, \theta; G) + L_{\text{KL}}^S(\phi; G) + L_{\text{unify}}(G) \quad (15)$$

$$L^H = L_{\text{recon}}^S(\psi, \theta; G) + L_{\text{KL}}^H(\psi; G) + L_{\text{unify}}(G) \quad (16)$$

## 3.4 Training

This section details **NMM-GNN**'s training framework, using non-Euclidean VAE, for representing social networks. We describe the optimization method for variables from Sections 3.2 and 3.3.

**Embedding Initialization.** We randomly initialize all embeddings of $z_i^S$ and $z_i^H$. For homophily regulated nodes, we choose a value for norm $w^S$, that is sampled from uniform distribution: $w^S : w^S \in [0, 1) \rightarrow \text{Unif}([0, 1))$ for all nodes, and for social influence regulated nodes, we choose a value for norm $w_{z_i}^H$, assigned uniformly at random per node. We set curvature values of spherical and hyperbolic spaces as $K_S = 1$ and $K_H = -1$ respectively. We leave the non-trivial problem of learning optimal curvatures as future work.

**Training procedure for homophily regulated nodes.** Parameter optimization for learning node embeddings is performed using Riemannian stochastic gradient descent (RSGD) for the spherical space as shown in Table 7. To ensure the updated node embeddings remain in norm-$w^S$ space, we perform a rescaling operation, $\text{proj}_S$, to project out-of-boundary embeddings back to the surface of

Table 7: Training Procedure for homophily regulated nodes and social influence regulated nodes.

(a) Homophily Regulated Nodes      (b) Social Influence Regulated Nodes

$$r(\boldsymbol{z}_{i,t}^S, L^S) = \big(1 + \frac{\boldsymbol{z}_{i,t}^{S^T}\nabla L^S(\boldsymbol{z}_{i,t}^S)}{\|\nabla L^S(\boldsymbol{z}_{i,t}^S)\|}\big)(I - \boldsymbol{z}_{i,t}^S \boldsymbol{z}_{i,t}^{S^T}) \quad (17)$$

$$\boldsymbol{z}_{i,t+1}^S \leftarrow \text{proj}_S\big(-\eta_t \cdot r(\boldsymbol{z}_{i,t}^S, L^S)\nabla L^S(\boldsymbol{z}_{i,t}^S)\big) \quad (18)$$

$$\text{SGD}^S(x) : x_{t+1} \leftarrow x_t - \eta_t \nabla L^S(x_t) \quad (19)$$

$$\boldsymbol{z}_{i,t+1}^H \leftarrow \boldsymbol{z}_{i,t}^H - \eta_t\big(\frac{1 - \|\boldsymbol{z}_{i,t}^H\|^2}{2}\big)^2 \nabla L^H(\boldsymbol{z}_{i,t}^H) \quad (20)$$

$$\text{SGD}^H(x) : x_{t+1} \leftarrow x_t - \eta_t \nabla L^H(x_t) \quad (21)$$

Table 8: Dataset statistics for evaluation datasets.

| Dataset | # Vertices | # Edges | Type | # Classes |
|---|---|---|---|---|
| BlogCatalog | 10.3K | 334.0K | undirected | 39 |
| LiveJournal | 4.8M | 69.0M | directed | 10 |
| Friendster | 65.6M | 1.8B | undirected | – |

the $w^S$-ball. We further update scalar parameters $J$ and $B$ (from the homophily regulated distribution) and $\beta, \lambda, a$ (from the spherical gaussian prior) through stochastic gradient descent (SGD) as defined below via $\text{SGD}^S(J), \text{SGD}^S(B), \text{SGD}^S(\beta), \text{SGD}^S(\lambda), \text{SGD}^S(a)$.

**Training procedure for social influence regulated nodes.** Parameter optimization for learning node embeddings is performed using RSGD for the hyperbolic space as shown in Table 7. The corresponding norm space, $w_{z_i}^H$, is also learned through RSGD by updating embeddings of $\boldsymbol{z}_i^H$. We further update scalar parameters $C$ and $D$ (from the social influence regulated distribution) and $\zeta$ (from the hyperbolic gaussian prior) through SGD as defined below via $\text{SGD}^H(C), \text{SGD}^H(D), \text{SGD}^H(\zeta)$.

**Training procedure for NMM-GNN weights.** Parameter optimization for $\gamma$ uses SGD as follows, where $L_{\text{total}} = L^S + L^H$:

$$\gamma_{t+1} \leftarrow \gamma_t - \eta_t \nabla L_{\text{total}}(\gamma_t) \quad (22)$$

## 4 Experiments

We comprehensively evaluate **NMM-GNN** on social network generation for popular large-scale social networks through multi-label classification and link prediction tasks against competitive SOTA baselines in various categories. The Appendix section further details Ablation Studies where we test quality of using (1) a mixture model, (2) distinct non-Euclidean geometric spaces, (3) non-Euclidean GNN-based encoders and non-Euclidean GraphVAE framework (through the inductive setting), and (4) space unification loss component. All experiments and each of the ablation studies consistently show **NMM-GNN** outperforms baseline network embedding models on all metrics for all datasets.

### 4.1 Datasets

For comprehensive evaluation, we assess our models on real-world datasets from well-known social media venues: *BlogCatalog (BC)* [27], *LiveJournal (LJ)* [28], and *Friendster (F)* [29] which are friendship networks among bloggers. Table 8 provides statistics of the datasets. In the Appendix, we also include experiments for Wikipedia datasets, to show that our model can also benefit other networks. Our research goal is to design an embedding model to better explain how the social network is formed. Thus, to separate model contribution from the learned representation, we focus on the setting of featureless graphs since quality of node features can be a confounding factor in determining quality of our model. However, since our model can handle feature graphs, we also provide these experiments in the Appendix, with our model outperforming all baselines on attributed networks.

### 4.2 Models

**Baselines.** We compare **NMM-GNN** to SOTA network embedding models, in Table 9, and report results in Table 10. We omit comparison to prior models e.g., **LINE** [30] due to lower performance. We also highlight that unlike prior works like $\kappa$-**GCN**, our model overcomes limitations of the product space, e.g., where the entire model belongs to a Cartesian product of non-Euclidean geometric spaces by default. Our work is in a category called *mixed space* model that uses a multi-geometric space framework where different portions of the graph may possibly belong to different spaces (based on the amount of impact each of homophily and social influence has for that personalized pair of

Table 9: Category and description of baseline models.

| Category | Description |
|---|---|
| Structural Embedding Models | **GraRep** [31], shallow embedding integrating global structural information |
| | **RolX** [32], unsupervised learning approach using structural role based similarity |
| | **GraphWave** [33], shallow embedding model using spectral graph wavelet diffusion patterns |
| GNN Embedding Models (Euclidean space) | **GraphSAGE** [34], inductive framework using node features and neighbor aggregation |
| | **GCN** [35], semi-supervised learning model via graph convolution on local neighborhoods |
| | **GAT** [36], graph attention model using mask self-attention layers on local neighborhoods |
| | **GIN** [37], graph embedding model based on the Weisfeiler-Lehman (WL) graph isomorphism test |
| | **GRAPHCL** [38], graph contrastive learning framework for unsupervised graph data |
| Homophily-based Embedding Models | **GELTOR** [17], embedding method using learning-to-rank with AdaSim* similarity metric |
| | **NRP** [27], embedding model using pairwise personalized PageRank on the global graph |
| GNN Embedding Models (non-Euclidean space) | **HGCN** [24], hyperbolic GCN model utilizing Riemannian geometry and hyperboloid model |
| | $\kappa$**-GCN** [39], GCN model using product space e.g., product of constant curvature spaces |
| Mixture Models (homophily and social influence) | **RaRE** [6], Bayesian probabilistic model for node proximity/popularity via posterior estimation |
| | **NMM**, our non-Euclidean mixture model (see Eqn. 6), without use of GraphVAE framework |
| | **NMM-GNN** , our non-Euclidean mixture model with non-Euclidean GraphVAE framework |

nodes). In the extreme case (Case 1) where only social influence is at play, e.g., weight of homophily representation is learned close to 0, the hyperbolic space will be used. On the other hand if only homophily is at play (Case 2), e.g., weight of homophily representation is learned close to 0, the spherical space will be used. In the normal case of both factors at play (Case 3), then both spaces will be used and can be jointly aligned with our space alignment mechanism. When using product space, Cases 1, 2, and 3 will all not be distinguished from each other as all cases will be modeled by one complex non-Euclidean geometric space as a Cartesian product of spherical and hyperbolic spaces.

**Time Complexity of NMM-GNN.** Regarding our model, **NMM** is highly efficient, with time complexity $O(ed + nd)$, where $n$ is number of nodes, $e$ is number of edges, and $d$ is dimension size. In comparison, the time complexity analyses for the remaining baseline models are as follows: the mixture model of **RaRE** is $O(ed + nd)$ which is comparable to the **NMM** mixture model, and the GNN embedding models of **GCN**, **GAT** (with one-head attention), and $\kappa$**-GCN** (for $\kappa = 0$) are $O(ed + nd^2)$. $\kappa$**-GCN** (for $\kappa \neq 0$) and **HGCN**'s time complexities are $O(ed + a \cdot nd^2)$, where $a$ is the filter length, and **NMM-GNN** is $O(ed + nd^2)$ which is comparable to GNN embedding models. As our work focuses on improving accuracy of learned embeddings, we further use GraphVAE training with **NMM** to achieve SOTA performance. We would like to point out that GraphVAE (of **NMM-GNN**) training is also designed to be highly parallelizable, which allows for scalability. Moreover, our model is capable of learning on real-world, highly large-scale graphs on the order of millions of nodes and billion of edges, e.g., Friendster, while achieving the best performance, which attests to its practical value to the network science community.

## 4.3 Evaluation

We detail our evaluation procedure for multi-label classification and link prediction. For all experiments, for fairness of comparison to baselines, we utilize the experiment procedure of [6]. Specifically, 90% of links are randomly sampled as training data. We do not perform cross-validation, since it may cause overfitting to occur as our framework uses learnable parameters e.g., $\mathbf{z}^S$, $\mathbf{z}^H$, $J$, $B$, $C$, $D$, $\gamma$, $\beta$, $\lambda$, $\alpha$, $\mathbf{W}_l$, and $\zeta$ which is a function of $\mathbf{z}^H$ equivalently interpreted as mean square error. Per dataset, we choose hyperparameter values for $\lambda_A$ in reconstruction loss: {0, 1, 2, 4, 8, 16, 32, 64}, step sizes $\eta_t$: {0.005, 0.001, 0.01, 0.05, 0.1}, and experiments are performed on AWS cluster (8 Nvidia GPUs).

### 4.3.1 Classification

Evaluation results are in Table 10. We observe that mixture models (homophily and social influence), achieve better performance on all datasets for all metrics. Specifically, it improves over structural embedding models (**GraphWave**), GNNs (**GAT**, **HGCN**), and homophily-based models (**GELTOR**, **NRP**). We also see learning embeddings in non-Euclidean geometric spaces helps better represent structures in social networks (**HGCN** vs. **GAT**, **RaRE** vs. **NMM**). We further observe using GNN-based encoders with GraphVAE learning yields additional improvement (**NMM** vs. **NMM-GNN**).

### 4.3.2 Link Prediction

As from [6], we measure quality of link prediction by sorting probability scores of every pair of nodes per model and evaluating them using area under the ROC curve (AUC) score. Specifically, 10% of existing edges and non-existing edges are hidden from training set, and probabilities are examined by

Table 10: Results of social network classification and link prediction for **Jaccard Index (JI)**, **Hamming Loss (HL)**, **F1 Score (F1)**, and **AUC** in % using embedding dimension 64. Our **NMM** and its variants are in gray shading. For each group of models, the best results are bold-faced. The overall best results on each dataset are underscored. †Ablation study variant models using distinct non-Euclidean geometric spaces for **NMM** (homophily/social influence) where $\mathbb{E}$, $\mathbb{S}$, and $\mathbb{H}$ denote Euclidean, Spherical, and Hyperbolic spaces.

| Datasets | BlogCatalog | | | | LiveJournal | | | | Friendster | | | |
| Metrics | JI | HL | F1 | AUC | JI | HL | F1 | AUC | JI | HL | F1 | AUC |
|---|---|---|---|---|---|---|---|---|---|---|---|---|
| **GraRep** | 36.0 | 28.2 | 45.6 | 87.9 | 40.1 | 41.1 | 35.2 | 56.7 | 53.6 | 34.2 | 40.6 | 89.8 |
| **RolX** | 37.2 | 25.4 | 48.7 | 90.4 | 40.9 | 38.0 | 35.6 | **60.1** | 58.8 | 33.9 | 40.9 | 90.3 |
| **GraphWave** | **39.5** | **22.8** | **48.9** | **92.3** | **42.2** | 37.6 | 35.9 | **60.1** | **59.0** | **31.5** | **41.1** | **90.5** |
| **GraphSAGE** | 45.4 | 20.1 | 49.3 | **92.0** | 45.5 | 34.7 | 34.1 | 59.0 | 64.1 | 28.7 | 43.4 | 90.5 |
| **GCN** | 47.3 | 19.5 | 55.1 | 91.6 | 46.7 | 31.2 | 47.8 | 62.6 | 66.5 | 28.0 | 47.2 | 91.9 |
| **GAT** | **47.9** | **19.3** | 54.5 | 91.4 | 47.4 | 28.5 | **49.0** | 65.3 | 66.3 | 28.0 | 46.8 | 92.0 |
| **GIN** | 47.1 | 19.7 | **56.2** | 91.5 | 48.6 | 28.3 | 48.1 | 67.2 | 66.0 | 27.7 | 48.1 | 92.3 |
| **GRAPHCL** | 47.5 | 19.4 | 55.8 | 91.3 | **49.7** | **27.9** | **49.0** | **69.4** | **68.1** | **25.5** | **49.9** | **92.8** |
| **GELTOR** | 47.4 | **19.3** | 54.9 | 92.0 | 51.0 | 28.9 | 48.6 | 65.3 | 66.7 | 27.9 | 47.5 | 91.7 |
| **NRP** | **61.6** | 20.4 | **65.2** | **95.5** | **69.7** | **24.5** | **64.0** | **78.7** | **72.2** | **22.6** | **52.8** | **92.2** |
| **HGCN** | 56.7 | **19.2** | 60.9 | 92.7 | 58.8 | **27.1** | **57.7** | 68.5 | **69.9** | 24.3 | 49.9 | **93.3** |
| $\kappa$-**GCN** | **61.6** | 20.7 | **65.4** | **95.3** | **63.6** | 27.3 | 57.2 | **69.1** | 69.4 | **24.1** | **50.3** | 93.1 |
| **RaRE** | 61.4 | 20.6 | 65.6 | 95.1 | 74.2 | 23.8 | 65.1 | 79.9 | 75.7 | 22.5 | 55.0 | 94.4 |
| **NMM**($\mathbb{H}^d/\mathbb{S}^d$)† | 56.6 | 19.8 | 62.3 | 95.1 | 74.0 | 28.4 | 55.5 | 68.8 | 74.6 | 26.9 | 50.6 | 93.0 |
| **NMM**($\mathbb{S}^d/\mathbb{S}^d$)† | 57.1 | 19.6 | 65.9 | 94.0 | 74.7 | 27.6 | 57.1 | 69.0 | 75.3 | 26.2 | 52.5 | 93.4 |
| **NMM**($\mathbb{E}^d/\mathbb{E}^d$)† | 57.9 | 19.5 | 66.3 | 95.4 | 75.1 | 25.0 | 58.4 | 71.2 | 77.0 | 24.7 | 52.8 | 94.5 |
| **NMM**($\mathbb{S}^d/\mathbb{E}^d$)† | 59.2 | 19.2 | 67.1 | 95.5 | 75.3 | 24.4 | 59.3 | 74.5 | 77.5 | 23.3 | 54.3 | 94.5 |
| **NMM**($\mathbb{H}^d/\mathbb{H}^d$)† | 58.4 | 19.0 | 66.7 | 95.3 | 75.6 | 24.6 | 61.9 | 76.0 | 78.8 | 23.3 | 55.0 | 94.7 |
| **NMM**($\mathbb{E}^d/\mathbb{H}^d$)† | 60.3 | 19.1 | 67.8 | 95.7 | 76.2 | 23.2 | 64.4 | 79.2 | 79.1 | 22.6 | 55.4 | 94.5 |
| **NMM** (ours) | **62.7** | 19.0 | 70.9 | 95.8 | 76.5 | 22.7 | **67.3** | 84.2 | 79.8 | 22.1 | 56.3 | 94.8 |
| **NMM-GNN** (ours) | 62.6 | **17.3** | **78.8** | **96.9** | **78.6** | **20.4** | **67.3** | **86.8** | **83.3** | **21.8** | **57.7** | **94.9** |

the model. Further, 10% of non-training edges are used for validation. For fairness against baselines on undirected networks, we treat all directed networks as undirected. Table 10 shows evaluation results for AUC score. Comparing relative score differences between best performing homophily embedding model (**NRP**) to **RaRE**, we observe that *LiveJournal* and *Friendster* datasets contain relatively more node social influence than *BlogCatalog*, which homophily-based models do not capture. Due to the above observation, it is likely that *LiveJournal* and *Friendster* (which are also larger datasets) show more realistic heterogeneity in network structure compared to *BlogCatalog* dataset e.g., cyclic structures produced by homophily based nodes and tree-like structures produced by social influence based nodes. Thus, modeling these structures in non-Euclidean spaces (**RaRE** vs. **NMM**) also shows more improvement. Moreover, in our **NMM** variant models, *every* node is influenced by *both* factors of homophily and social influence through our non-Euclidean mixture model. It is a *weighted combination* personalized per node of these factors that influence the links formed, rather than being generated solely through homophily vs. social influence.

## 5 Conclusions

We are among the first to explore a Graph-based non-Euclidean mixture model for social networks. As social networks are influenced by homophily and social influence, we design a model to represent both factors jointly for nodes. Further, we model resulting unique network topologies (cycles and trees) using distinct non-Euclidean geometric spaces and introduce a GNN-based non-Euclidean variational autoencoder framework for our model, to effectively learn embeddings. The resulting model, **NMM-GNN**, significantly outperforms various state-of-the-art models for social networks. As future work, we hope to explore alternatives to graph-based VAE methods for improved learning.

# 6 Acknowledgments and Disclosure of Funding

This work was partially supported by DARPA HR0011-24-9-0370; NSF 1937599, 2106859, 2119643, 2200274, 2211557, 2303037, 2312501; NIH U54HG012517, U24DK097771; Optum AI; NASA; SRC JUMP 2.0 Center; Amazon Research Awards; and Snapchat Gifts.

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

# Appendix

## A    Limitations

**NMM** is highly efficient, with time complexity $O(ed + nd)$, where $n$ is number of nodes, $e$ is number of edges, and $d$ is dimension size. As our work focuses on improving accuracy of learned embeddings, we further use GraphVAE training with **NMM** to achieve SOTA performance, called **NMM-GNN**. Using the GraphVAE training pipeline may be seen as a limitation as it is less efficient compared to GNNs for training time. However, in the worst case scenario, **NMM-GNN** still achieves efficiency comparable to popular heterogeneous graph models including GraphRNN [40], and training is designed to be highly parallelizable, which allows for scalability. Moreover, our model is capable of learning on real-world, highly large-scale graphs on the order of millions of nodes and billion of edges, e.g., *Friendster*, while achieving the SOTA performance, which attests to its practical value to the network science community.

Our model also makes the assumption the homophily regulated nodes lie on the surface of the spherical ball and the social influence regulated nodes lie on the open Poincare ball, which are both natural representations for modeling nodes. People have observed that more popular celebrity nodes tend to have smaller norm space and are embedded towards the center of the ball (similar to higher order concepts in the knowledge graph space), while less popular nodes (containing less social influence) are embedded towards the boundary of the ball (similar to entities in the knowledge graph space). To ensure that we can compute an intersection space (for the space unification component to unify the homophily and social influence representation for the same node), we enforce that the spherical surface norm is in the Poincare ball e.g., any learned soft value norm space between 0 and 1. In compute, while we evaluate on a cluster of 8 GPUs, at least one GPU is necessary for running our experiments (which may been seen as compute limitation).

Regarding ethical considerations and fairness, privacy and fairness are often topics of focus regarding social networks. Our model uses information like node popularity based on graph structure (in degree vs. out degree ratio), as well as attributed features if available. Further it also infers links through neighborhood context by looking at connected nodes. In general, a network embedding model uses personal information from the user which may impinge on their privacy. However, we make every attempt in our model to allow the user to select their granular choice of model personalization (e.g., we provide the option to choose from whether or not users want to enable their attributed information like profile interest and history being learned).

## B    A Note on Graph-Level Learning

Our work can be generalized to the graph-level because our method learns to represent social science network factors based on topologies in the graph on clusters of nodes and edges. Thus, if the cluster of nodes and edges comprised of the entire graph and we subsequently applied graph pooling per node embedding (that we currently learn), we can reduce the embedding from node level to graph level. In this way, homophily and social influence can be modeled at the graph level. That said, it is unclear whether graph-level modeling would be specifically useful or interpretable for social network embedding models as compared to node-level modeling. This is due to the social network setting requiring links to be generated that are per node and not at the graph level, because a user (modeled as a node) is recommended to another specific user in the practical social network setting. This is different from other network domains like molecular classification where the entire graph represents one molecule e.g., atoms form individual nodes and chemical bonds form the edges. For this reason, node-level learning is in fact consistent with the recent state-of-the-art NN methods in the network science community though **NMM-GNN** can still be generalized to learning at the graph-level.

## C    Additional Experiments

In this section, we provide experiment results on Wikipedia networks and on attributed graphs. As in the case for social networks, we also evaluate on multi-label classification and link prediction for Wikipedia networks and attributed graphs. For all experiments, for fairness of comparison to baselines, we utilize experiment procedure of [6]. Specifically, 90% of links are randomly sampled

as training data. We do not perform cross-validation, since it may cause overfitting to occur as our framework uses learnable parameters e.g., $\mathbf{z}^S$, $\mathbf{z}^H$, $J$, $B$, $C$, $D$, $\gamma$, $\beta$, $\lambda$, $\alpha$, $\mathbf{W}_l$, and $\zeta$ which is a function of $\mathbf{z}^H$ equivalently interpreted as mean square error. Per dataset, we choose hyperparameter values for $\lambda_A$ in reconstruction loss: $\{0, 1, 2, 4, 8, 16, 32, 64\}$, and step sizes $\eta_t$: $\{0.005, 0.001, 0.01, 0.05, 0.1\}$.

**Evaluation on Wikipedia networks.** In Table 12, we additionally present multi-label classification and link prediction experiments on Wikipedia datasets to show the benefits of our model on other networks. These include *Wikipedia Clickstream* [6], which contains counts of (referrer, resource) pairs extracted from the request logs of Wikipedia, and *Wikipedia Hyperlink* [6], which contains edges as hyperlinks from one page to another. Dataset statistics for the Wikipedia datasets are summarized in Table 11. Our **NMM** model variants consistently achieves the best performance over all the competitive baseline models belonging to categories of (1) structural embedding models, (2) GNN embedding models (Euclidean space), (3) homophily-based embedding models, (4) GNN embedding models (non-Euclidean space), and (5) mixture models. This shows that **NMM**'s model is generalizable and widely applicable because it can learn effective representations on online information networks that go beyond social networks.

Table 11: Dataset statistics for evaluation datasets.

| Dataset | # Vertices | # Edges | Type | # Classes |
|---|---|---|---|---|
| Wikipedia Clickstream | 2.4M | 15.0M | directed | 6 |
| Wikipedia Hyperlink | 488K | 5.5M | directed | 6 |

Table 12: Results of social network classification and link prediction for **Jaccard Index (JI)**, **Hamming Loss (HL)**, **F1 Score (F1)**, and **AUC** in % using embedding dimension 64. Our **NMM** and variants are in gray shading. For each group of models, best results are bold-faced. The overall best results on each dataset are underscored.[†]Ablation study variant models using distinct non-Euclidean geometric spaces for **NMM** (homophily/social influence) where $\mathbb{E}$, $\mathbb{S}$, and $\mathbb{H}$ denote Euclidean, Spherical, and Hyperbolic spaces.

| Datasets | Wikipedia Clickstream | | | | Wikipedia Hyperlink | | | |
|---|---|---|---|---|---|---|---|---|
| Metrics | JI | HL | F1 | AUC | JI | HL | F1 | AUC |
| **GraRep** | 31.9 | 28.3 | 44.4 | 79.6 | 48.3 | 22.7 | 50.2 | 76.8 |
| **RolX** | 32.2 | 28.1 | **44.9** | 85.4 | **57.1** | **17.6** | **56.0** | **82.6** |
| **GraphWave** | **32.8** | **27.0** | 44.6 | **86.1** | 55.3 | 18.1 | 54.8 | 81.8 |
| **GraphSAGE** | 33.1 | 26.6 | 45.1 | 89.3 | 62.8 | 8.3 | 68.7 | 89.4 |
| **GCN** | 33.1 | **24.1** | 50.7 | 89.5 | 63.7 | **6.5** | **76.4** | 92.0 |
| **GAT** | 33.4 | 24.2 | **51.2** | 89.0 | 64.2 | **6.5** | 76.2 | 92.1 |
| **GIN** | **33.7** | 24.4 | 50.6 | **91.2** | **65.1** | 6.9 | 76.1 | 92.3 |
| **GRAPHCL** | 33.3 | **24.1** | 51.0 | 90.9 | 64.5 | 6.6 | **76.4** | **92.7** |
| **GELTOR** | 33.2 | 24.1 | 50.9 | 91.4 | 64.0 | **6.8** | 76.7 | 92.3 |
| **NRP** | **46.5** | **20.9** | **57.1** | **91.8** | **75.5** | 7.1 | **81.1** | **96.9** |
| **HGCN** | **38.1** | **20.6** | 54.3 | 89.9 | 69.8 | 6.0 | 78.4 | 94.1 |
| **$\kappa$-GCN** | 37.6 | 20.9 | **55.0** | **90.2** | **75.0** | 7.1 | **81.6** | **96.9** |
| **RaRE** | 47.8 | 20.7 | 58.0 | 93.0 | 75.7 | 6.8 | 82.3 | 97.5 |
| **NMM$(\mathbb{H}^d/\mathbb{S}^d)^†$** | 44.8 | 27.8 | 44.2 | 86.5 | 76.0 | 8.2 | 76.4 | 92.1 |
| **NMM$(\mathbb{S}^d/\mathbb{S}^d)^†$** | 45.1 | 22.9 | 55.7 | 90.3 | 78.4 | 7.1 | 79.7 | 97.0 |
| **NMM$(\mathbb{E}^d/\mathbb{E}^d)^†$** | 46.2 | 21.7 | 56.0 | 90.6 | 79.2 | 6.8 | 82.0 | 97.5 |
| **NMM$(\mathbb{S}^d/\mathbb{E}^d)^†$** | 47.7 | 20.9 | 56.2 | 91.0 | 79.9 | 6.6 | 82.3 | 97.6 |
| **NMM$(\mathbb{H}^d/\mathbb{H}^d)^†$** | 47.4 | 20.9 | 56.8 | 91.2 | 81.0 | 6.6 | 81.6 | 97.3 |
| **NMM$(\mathbb{E}^d/\mathbb{H}^d)^†$** | 48.1 | 20.7 | 57.9 | 91.7 | 80.5 | 6.2 | 82.4 | 97.8 |
| **NMM** (ours) | 49.2 | 18.5 | 59.0 | 92.8 | 80.3 | 5.8 | 82.5 | **98.0** |
| **NMM-GNN** (ours) | **49.7** | **16.5** | **60.8** | **95.6** | **81.7** | **5.5** | **83.0** | 97.3 |

**Evaluation on attributed graphs.** As our model can also handle attributed graphs, we provide experiments against SOTA attributed network embedding models that are summarized below. Since these models require the presence of network attributes, we do not include them in our experiments on featureless graphs for fairness of comparison. We conduct evaluation on well-known large-scale attributed graph datasets which include *Facebook* [9] and *Google+* [9] social networks where consistent with [27], we treat each ego-network as a label and extract attributes from their user profiles. Dataset statistics are in Table 13, and results are reported in Table 14. It can be seen that our **NMM-GNN** model consistently achieves the best performance in both the large-scale graph datasets, indicating that our model's inherent learning ability is effective to learn *graph structure* and *topology*, which goes beyond simply exploiting information from node and edge attributes. Below is a summary of the baseline attributed network embedding models:

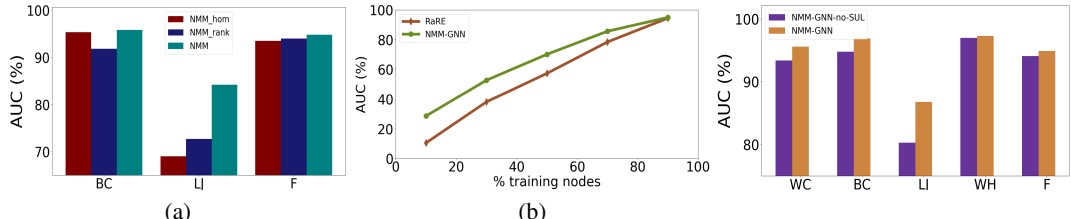

Figure 2: Ablation studies. (a) Quality of mixture model, where $\mathbf{NMM_{hom}}$ and $\mathbf{NMM_{rank}}$ are homophily-only and social influence-only deconstructed $\mathbf{NMM}$ components. (b) Inductive reasoning for $\mathbf{NMM\text{-}GNN}$ and $\mathbf{RaRE}$ on LiveJournal. *% nodes* $\{10, 30, 50, 70, 90\}$ are sampled ensuring no overlap with test. (c) Ablation study on quality of using space unification loss (SUL) component.

- **PANE** [41], random-walk based attirubted network embedding (ANE) model
- **NRP** [27], described in Table 9 of the main paper
- **CONN** [42], GNN for attributed networks via collaborative aggregation on bipartite graphs

Table 13: Dataset statistics for evaluation datasets.

| Dataset | # Vertices | # Edges | #Attributes | Type | # Classes |
|---|---|---|---|---|---|
| Facebook | 4.0K | 88.2K | 1.3K | undirected | 193 |
| Google+ | 107.6K | 13.7M | 15.9K | directed | 468 |

Table 14: Results of social network classification and link prediction for **Jaccard Index (JI)**, **Hamming Loss (HL)**, **F1 Score (F1)**, and **AUC** in % using embedding dimension 64. Our **NMM** and variants are in gray shading. The overall best results on each dataset are bold-faced.

| Datasets | Facebook | | | | Google+ | | | |
|---|---|---|---|---|---|---|---|---|
| Metrics | JI | HL | F1 | AUC | JI | HL | F1 | AUC |
| **NRP** | 48.0 | 19.7 | 55.8 | 96.1 | 40.7 | 26.5 | 51.1 | 81.3 |
| **PANE** | 49.3 | 15.9 | 64.3 | 96.4 | 45.1 | 23.8 | 60.2 | 92.2 |
| **CONN** | 51.2 | 16.4 | 61.2 | **96.8** | 44.9 | 23.3 | 61.3 | 90.8 |
| **NMM-GNN** (ours) | **52.6** | **14.2** | **67.1** | **96.8** | **47.3** | **20.4** | **66.0** | **93.9** |

The source code and datasets for our work can be found at: https://github.com/roshnigiyer/nmm.

# D    Ablation Studies

This section details ablation studies where we test quality of using (1) a mixture model, (2) distinct non-Euclidean geometric spaces, (3) non-Euclidean GNN-based encoders and non-Euclidean GraphVAE framework (through the inductive setting), and (4) space unification loss component. Each of the ablation studies provides insight to motivate our architecture design choices, as well as consistently shows **NMM-GNN** outperforms baseline network embedding models.

***Quality of using a mixture model architecture.***   We deconstruct our mixture model of **NMM**, to observe the effect different network factors have on learning embeddings, where $\alpha$ and $\beta$ are learnable. We report results on embedding dimension 64, evaluated on AUC score, with the models summarized below:

- $\mathbf{NMM_{hom}}$, deconstructed homophily component: $p_\theta(e_{ij} = 1) = \alpha \cdot p_{\mathrm{hom}}(e_{ij} = 1)$
- $\mathbf{NMM_{rank}}$, deconstructed social influence component: $p_\theta(e_{ij} = 1) = \beta \cdot p_{\mathrm{rank}}(e_{ij} = 1)$
- **NMM**, which is our mixture model defined by Eqn. 6

As shown in Figure 2(a), the mixture model of **NMM**  outperforms that of its subcomponents $\mathbf{NMM_{hom}}$  and $\mathbf{NMM_{rank}}$  on all datasets for link prediction. This validates the effectiveness of using a mixture model architecture for modeling both homophily and social influence factors jointly.

***Quality of using distinct non-Euclidean geometric spaces.*** We study combinations of geometric spaces to model **NMM** (homophily/social influence) to observe the effect it has on learning topological structure, denoted with $\textbf{NMM}(\cdot)^{\dagger}$ in Table 10 (main paper). As shown, the choice of modeling homophily based nodes in spherical space and modeling social influence based nodes in hyperbolic space leads to the best performance. Further, **NMM** outperforms its Euclidean space counterpart showing that the social network exhibits structures (cycles and hierarchy) that need to be appropriately represented in curved geometric spaces. There is also evidence of non-Euclidean topologies in the datasets. For example, the average node on LiveJournal has in-degree of 17 but outdegree of 25, showing several hierarchical structures present, and 17.7% of LiveJournal data contains cycles [43].

***Link prediction on unseen nodes (inductive task).*** We study ability of **NMM-GNN** to learn on the inductive setting (in addition to the standard transductive setting of Table 10 of the main paper). To do so, we randomly sample % of *nodes* being $\{10, 30, 50, 70, 90\}$ and their corresponding links as our training set. Test nodes are ensured to have no overlap with training nodes, to allow for link prediction on unseen graphs. Figure 2(b) reports results on **NMM-GNN** and **RaRE** for *LiveJournal* on embedding dimension of 64 for AUC score. **NMM-GNN** outperforms **RaRE** on all settings of training nodes. This shows the effectiveness of **NMM-GNN** in using non-Euclidean GNN-based encoders and a non-Euclidean GraphVAE training framework during the learning process, two components that **RaRE** lacks. Further, as less training nodes are observed, **NMM-GNN** outperforms **RaRE** by larger margins (e.g., 10% vs. 70% training nodes), showing **NMM-GNN** better generalizes to unseen graphs.

***Quality of using space unification loss.*** We test quality of using our proposed space unification loss (SUL) in our **NMM-GNN**, by conducting experiments for with the loss component (**NMM-GNN**) and without it (**NMM-GNN-no-SUL**). We report results on embedding dimension 64, evaluated on AUC score in Figure 2(c), for datasets *Wikipedia Clickstream (WC)*, *BlogCatalog (BC)*, *LiveJournal (LJ)*, *Wikipedia Hyperlink (WH)*, and *Friendster (F)*. As shown, using SUL improves performance on all datasets, indicating importance of bridging together representations of distinct non-Euclidean spaces (spherical and hyperbolic space) at node level.

# NeurIPS Paper Checklist

1. **Claims**

   Question: Do the main claims made in the abstract and introduction accurately reflect the paper's contributions and scope?

   Answer: [Yes]

   Justification: The abstract summarizes the main paper claims and we also provide a contribution summary with bullet points in the Introduction. Further, all these main claims of the paper are supported through Section 3 (detailing our model architecture), Section 4 on Experiments, as well as our (four) ablation studies in the Appendix.

   Our model novelly represents both homophily and social influence factors when modeling the social network, an advancement over the recent network embedding methods. Further, the novel utilization of non-Euclidean geometric spaces to model the resulting topologies due to network factors through appropriate positive and negative curvatures naturally exibiting properties of cycles and hierarchy (the observed topological heterogeneity), tremendously improves the representation capability of network embedding modes. This is evidenced by the empirical results as ablation studies for our work. Moreover, **NMM-GNN** not only improves against SOTA models in the performance metrics (Jaccard Index, Hamming Loss, , F1 score, AUC Score), but is also applicable to the Inductive Setting, other large-scale information networks like Wikipedia networks, and attributed networks (Facebook and Google+), while consistently achieving the best performance indicating model generalizability.

   Guidelines:
   - The answer NA means that the abstract and introduction do not include the claims made in the paper.
   - The abstract and/or introduction should clearly state the claims made, including the contributions made in the paper and important assumptions and limitations. A No or NA answer to this question will not be perceived well by the reviewers.
   - The claims made should match theoretical and experimental results, and reflect how much the results can be expected to generalize to other settings.
   - It is fine to include aspirational goals as motivation as long as it is clear that these goals are not attained by the paper.

2. **Limitations**

   Question: Does the paper discuss the limitations of the work performed by the authors?

   Answer: [Yes]

   Justification: We have an entire section on Limitations in the Appendix (Section A) that addresses all the guidelines below. Our model also makes the assumption the homophily regulated nodes lie on the surface of the spherical ball and the social influence regulated nodes lie on the open Poincare ball, which are both natural representations for modeling nodes. People have observed that more popular celebrity nodes tend to have smaller norm space and are embedded towards the center of the ball (similar to higher order concepts in the knowledge graph space), while less popular nodes (containing less social influence) are embedded towards the boundary of the ball (similar to entities in the knowledge graph space). To ensure that we can compute an intersection space (for the space unification component to unify the homophily and social influence representation for the same node), we enforce that the spherical surface norm is in the Poincare ball e.g., any learned soft value norm space between 0 and 1. In compute, while we evaluate on a cluster of 8 GPUs, at least one GPU is necessary for running our experiments (which may been seen as compute limitation).

   Regarding ethical considerations and fairness, privacy and fairness are often topics of focus regarding social networks. Our model uses information like node popularity based on graph structure (in degree vs. out degree ratio), as well as attributed features if available. Further it also infers links through neighborhood context by looking at connected nodes. In general, a network embedding model uses personal information from the user which may impinge on their privacy. However, we make every attempt in our model to allow the user to select their granular choice of model personalization (e.g., we provide the option to choose from

whether or not users want to enable their attributed information like profile interest and history being learned).

Guidelines:

- The answer NA means that the paper has no limitation while the answer No means that the paper has limitations, but those are not discussed in the paper.
- The authors are encouraged to create a separate "Limitations" section in their paper.
- The paper should point out any strong assumptions and how robust the results are to violations of these assumptions (e.g., independence assumptions, noiseless settings, model well-specification, asymptotic approximations only holding locally). The authors should reflect on how these assumptions might be violated in practice and what the implications would be.
- The authors should reflect on the scope of the claims made, e.g., if the approach was only tested on a few datasets or with a few runs. In general, empirical results often depend on implicit assumptions, which should be articulated.
- The authors should reflect on the factors that influence the performance of the approach. For example, a facial recognition algorithm may perform poorly when image resolution is low or images are taken in low lighting. Or a speech-to-text system might not be used reliably to provide closed captions for online lectures because it fails to handle technical jargon.
- The authors should discuss the computational efficiency of the proposed algorithms and how they scale with dataset size.
- If applicable, the authors should discuss possible limitations of their approach to address problems of privacy and fairness.
- While the authors might fear that complete honesty about limitations might be used by reviewers as grounds for rejection, a worse outcome might be that reviewers discover limitations that aren't acknowledged in the paper. The authors should use their best judgment and recognize that individual actions in favor of transparency play an important role in developing norms that preserve the integrity of the community. Reviewers will be specifically instructed to not penalize honesty concerning limitations.

3. **Theory Assumptions and Proofs**

Question: For each theoretical result, does the paper provide the full set of assumptions and a complete (and correct) proof?

Answer: [Yes]

Justification: The assumptions about the social network graph as well as our model are comprehensively listed in Section 3 which contains the "Methodology" and "Overview" assumptions as well as subsections that each list the associated assumptions for that particular model component. Moreover, Section 4 (Experiments) and Appendix (Additional Experiments + Ablation Studies) detail the entire experiment evaluation procedure and design as well as all the corresponding graph assumptions.

Guidelines:

- The answer NA means that the paper does not include theoretical results.
- All the theorems, formulas, and proofs in the paper should be numbered and cross-referenced.
- All assumptions should be clearly stated or referenced in the statement of any theorems.
- The proofs can either appear in the main paper or the supplemental material, but if they appear in the supplemental material, the authors are encouraged to provide a short proof sketch to provide intuition.
- Inversely, any informal proof provided in the core of the paper should be complemented by formal proofs provided in appendix or supplemental material.
- Theorems and Lemmas that the proof relies upon should be properly referenced.

4. **Experimental Result Reproducibility**

Question: Does the paper fully disclose all the information needed to reproduce the main experimental results of the paper to the extent that it affects the main claims and/or conclusions of the paper (regardless of whether the code and data are provided or not)?

Answer: [Yes]

Justification: The full details of experiments for train and test sampling and evaluation etc. as well as the comprehensive experiment design procedure are provided in Section 4 (Experiments). We also provide all hyperparameter setting details. Further, we provide all the details for each of the baseline models evaluated for both Experiments as well as Ablation Studies. Moreover, we release all source code and datasets used in the paper for enhanced reproducibility.

Guidelines:

- The answer NA means that the paper does not include experiments.
- If the paper includes experiments, a No answer to this question will not be perceived well by the reviewers: Making the paper reproducible is important, regardless of whether the code and data are provided or not.
- If the contribution is a dataset and/or model, the authors should describe the steps taken to make their results reproducible or verifiable.
- Depending on the contribution, reproducibility can be accomplished in various ways. For example, if the contribution is a novel architecture, describing the architecture fully might suffice, or if the contribution is a specific model and empirical evaluation, it may be necessary to either make it possible for others to replicate the model with the same dataset, or provide access to the model. In general. releasing code and data is often one good way to accomplish this, but reproducibility can also be provided via detailed instructions for how to replicate the results, access to a hosted model (e.g., in the case of a large language model), releasing of a model checkpoint, or other means that are appropriate to the research performed.
- While NeurIPS does not require releasing code, the conference does require all submissions to provide some reasonable avenue for reproducibility, which may depend on the nature of the contribution. For example
  (a) If the contribution is primarily a new algorithm, the paper should make it clear how to reproduce that algorithm.
  (b) If the contribution is primarily a new model architecture, the paper should describe the architecture clearly and fully.
  (c) If the contribution is a new model (e.g., a large language model), then there should either be a way to access this model for reproducing the results or a way to reproduce the model (e.g., with an open-source dataset or instructions for how to construct the dataset).
  (d) We recognize that reproducibility may be tricky in some cases, in which case authors are welcome to describe the particular way they provide for reproducibility. In the case of closed-source models, it may be that access to the model is limited in some way (e.g., to registered users), but it should be possible for other researchers to have some path to reproducing or verifying the results.

5. **Open access to data and code**

   Question: Does the paper provide open access to the data and code, with sufficient instructions to faithfully reproduce the main experimental results, as described in supplemental material?

   Answer: [Yes]

   Justification: We provide open access to all our source code as well as for the datasets in the Supplemental Material that has been uploaded. As part of this, we also include a README file to details the setup/requirement as well as commands instructions. We also refer to this in Section 3 (Methodology) and Section 4 (Experiments). In addition, we detail all the hyperparameter settings in the main paper (Section 4).

   Guidelines:

   - The answer NA means that paper does not include experiments requiring code.
   - Please see the NeurIPS code and data submission guidelines (https://nips.cc/public/guides/CodeSubmissionPolicy) for more details.

- While we encourage the release of code and data, we understand that this might not be possible, so "No" is an acceptable answer. Papers cannot be rejected simply for not including code, unless this is central to the contribution (e.g., for a new open-source benchmark).
- The instructions should contain the exact command and environment needed to run to reproduce the results. See the NeurIPS code and data submission guidelines (https://nips.cc/public/guides/CodeSubmissionPolicy) for more details.
- The authors should provide instructions on data access and preparation, including how to access the raw data, preprocessed data, intermediate data, and generated data, etc.
- The authors should provide scripts to reproduce all experimental results for the new proposed method and baselines. If only a subset of experiments are reproducible, they should state which ones are omitted from the script and why.
- At submission time, to preserve anonymity, the authors should release anonymized versions (if applicable).
- Providing as much information as possible in supplemental material (appended to the paper) is recommended, but including URLs to data and code is permitted.

6. **Experimental Setting/Details**

   Question: Does the paper specify all the training and test details (e.g., data splits, hyper-parameters, how they were chosen, type of optimizer, etc.) necessary to understand the results?

   Answer: [Yes]

   Justification: We have an entire training procedure section and loss function information inside Section 3 (Methodology) which comprehensively describes our model architecture. All optimizer and loss function details are presented here. Our Section 4 (Experiments) and Appendix (which contains our four Ablation Studies) detail the experiment evaluation for the corresponding experiments and detail all the training and test details as well as any additional information needed to understand the results. We provide detailed information about hyperparameter settings (Section 4), which were selected based on performance on the validation set (train/validation/test split details also in this section). Further, the use of all evaluation metrics from the experiments are clearly defined (paper resource references pointed) and justified.

   Guidelines:
   - The answer NA means that the paper does not include experiments.
   - The experimental setting should be presented in the core of the paper to a level of detail that is necessary to appreciate the results and make sense of them.
   - The full details can be provided either with the code, in appendix, or as supplemental material.

7. **Experiment Statistical Significance**

   Question: Does the paper report error bars suitably and correctly defined or other appropriate information about the statistical significance of the experiments?

   Answer: [NA]

   Justification: We provide all details about distribution priors, random embedding initialization etc., and evaluate our model on standard experiment metrics important to the social network community for determining model quality. Error bars are not applicable in this scenario but our paper details all experiment conditions.

   Guidelines:
   - The answer NA means that the paper does not include experiments.
   - The authors should answer "Yes" if the results are accompanied by error bars, confidence intervals, or statistical significance tests, at least for the experiments that support the main claims of the paper.
   - The factors of variability that the error bars are capturing should be clearly stated (for example, train/test split, initialization, random drawing of some parameter, or overall run with given experimental conditions).

- The method for calculating the error bars should be explained (closed form formula, call to a library function, bootstrap, etc.)
- The assumptions made should be given (e.g., Normally distributed errors).
- It should be clear whether the error bar is the standard deviation or the standard error of the mean.
- It is OK to report 1-sigma error bars, but one should state it. The authors should preferably report a 2-sigma error bar than state that they have a 96% CI, if the hypothesis of Normality of errors is not verified.
- For asymmetric distributions, the authors should be careful not to show in tables or figures symmetric error bars that would yield results that are out of range (e.g. negative error rates).
- If error bars are reported in tables or plots, The authors should explain in the text how they were calculated and reference the corresponding figures or tables in the text.

8. **Experiments Compute Resources**

Question: For each experiment, does the paper provide sufficient information on the computer resources (type of compute workers, memory, time of execution) needed to reproduce the experiments?

Answer: [Yes]

Justification: The paper has a section devoted to analyzing the memory complexity of the model (Section 4: Experiments), as well as also provide compute and hyperparameter details in this section and Section 3 (model architecture). Additionally, we release all of our source code and datasets for maximal reproducibility (Supplement Material).

Guidelines:

- The answer NA means that the paper does not include experiments.
- The paper should indicate the type of compute workers CPU or GPU, internal cluster, or cloud provider, including relevant memory and storage.
- The paper should provide the amount of compute required for each of the individual experimental runs as well as estimate the total compute.
- The paper should disclose whether the full research project required more compute than the experiments reported in the paper (e.g., preliminary or failed experiments that didn't make it into the paper).

9. **Code Of Ethics**

Question: Does the research conducted in the paper conform, in every respect, with the NeurIPS Code of Ethics https://neurips.cc/public/EthicsGuidelines?

Answer: [Yes]

Justification: All paper guidelines have been carefully followed regarding page limit, formatting, content, research topic and relevance, writing, figures etc.

All privacy and data security concerns are also addressed as we utilize public benchmark datasets that do not contain any sensitive user information. Our Limitations section further provides details regarding ethical considerations.

Guidelines:

- The answer NA means that the authors have not reviewed the NeurIPS Code of Ethics.
- If the authors answer No, they should explain the special circumstances that require a deviation from the Code of Ethics.
- The authors should make sure to preserve anonymity (e.g., if there is a special consideration due to laws or regulations in their jurisdiction).

10. **Broader Impacts**

Question: Does the paper discuss both potential positive societal impacts and negative societal impacts of the work performed?

Answer: [Yes]

Justification: The paper provides clear motivation of the importance of the research problem of investigation and further identifies key challenges in the network science community in state-of-the-art embedding models, which thus motivates our model architecture. Further, our paper provides clear intuition and justification for each design choice decision in our model components. Additionally, impact of our model is discussed with respect to the social network community and more broadly for other online information networks e.g., Wikipedia datasets and attributed networks (in the Appendix section on Ablation studeies). Lastly, we also discuss Limitations of our model in Appendix Section A.

This model effectively also learns without needed data attributes, which could beneficially influence privacy considerations in social networks (that users do not need to have their profile information and history being stored). Of course, any network science model could have potential for misuse in areas like surveillance or manipulation of online communities, so litigation and action should be taken to safeguard against misuse.

Guidelines:

- The answer NA means that there is no societal impact of the work performed.
- If the authors answer NA or No, they should explain why their work has no societal impact or why the paper does not address societal impact.
- Examples of negative societal impacts include potential malicious or unintended uses (e.g., disinformation, generating fake profiles, surveillance), fairness considerations (e.g., deployment of technologies that could make decisions that unfairly impact specific groups), privacy considerations, and security considerations.
- The conference expects that many papers will be foundational research and not tied to particular applications, let alone deployments. However, if there is a direct path to any negative applications, the authors should point it out. For example, it is legitimate to point out that an improvement in the quality of generative models could be used to generate deepfakes for disinformation. On the other hand, it is not needed to point out that a generic algorithm for optimizing neural networks could enable people to train models that generate Deepfakes faster.
- The authors should consider possible harms that could arise when the technology is being used as intended and functioning correctly, harms that could arise when the technology is being used as intended but gives incorrect results, and harms following from (intentional or unintentional) misuse of the technology.
- If there are negative societal impacts, the authors could also discuss possible mitigation strategies (e.g., gated release of models, providing defenses in addition to attacks, mechanisms for monitoring misuse, mechanisms to monitor how a system learns from feedback over time, improving the efficiency and accessibility of ML).

11. **Safeguards**

Question: Does the paper describe safeguards that have been put in place for responsible release of data or models that have a high risk for misuse (e.g., pretrained language models, image generators, or scraped datasets)?

Answer: [NA]

Justification: We have been extremely careful on the type of data and model we use and to consider all ethical aspects respectively. No human subjects were used in our research study (N/A), and all of our datasets for evaluation are large-scale public benchmark popular social media datasets evaluated on numerous papers in the recent years. As such, these datasets are highly suitable for direct comparisons against various SOTA models as they have been widely tested on.

Guidelines:

- The answer NA means that the paper poses no such risks.
- Released models that have a high risk for misuse or dual-use should be released with necessary safeguards to allow for controlled use of the model, for example by requiring that users adhere to usage guidelines or restrictions to access the model or implementing safety filters.
- Datasets that have been scraped from the Internet could pose safety risks. The authors should describe how they avoided releasing unsafe images.

- We recognize that providing effective safeguards is challenging, and many papers do not require this, but we encourage authors to take this into account and make a best faith effort.

12. **Licenses for existing assets**

    Question: Are the creators or original owners of assets (e.g., code, data, models), used in the paper, properly credited and are the license and terms of use explicitly mentioned and properly respected?

    Answer: [Yes]

    Justification: All source codes are attributed to the respective authors and we even identify in our paper the training procedures if used from prior work for fairness of comparison e.g., **RaRE** [6] that we indicate in Section 4 (Experiments). We also release all our source code and datasets (in addition to referencing each of them in our main paper for the source information) for reproducibility under MIT and CC-by licences.

    Guidelines:

    - The answer NA means that the paper does not use existing assets.
    - The authors should cite the original paper that produced the code package or dataset.
    - The authors should state which version of the asset is used and, if possible, include a URL.
    - The name of the license (e.g., CC-BY 4.0) should be included for each asset.
    - For scraped data from a particular source (e.g., website), the copyright and terms of service of that source should be provided.
    - If assets are released, the license, copyright information, and terms of use in the package should be provided. For popular datasets, paperswithcode.com/datasets has curated licenses for some datasets. Their licensing guide can help determine the license of a dataset.
    - For existing datasets that are re-packaged, both the original license and the license of the derived asset (if it has changed) should be provided.
    - If this information is not available online, the authors are encouraged to reach out to the asset's creators.

13. **New Assets**

    Question: Are new assets introduced in the paper well documented and is the documentation provided alongside the assets?

    Answer: [NA]

    Justification: The paper is about improved model development for enhancing representation learning of nodes in the social network to better explain how links in the social network are formed. No new assets or datasets etc. are released as part of this work.

    Guidelines:

    - The answer NA means that the paper does not release new assets.
    - Researchers should communicate the details of the dataset/code/model as part of their submissions via structured templates. This includes details about training, license, limitations, etc.
    - The paper should discuss whether and how consent was obtained from people whose asset is used.
    - At submission time, remember to anonymize your assets (if applicable). You can either create an anonymized URL or include an anonymized zip file.

14. **Crowdsourcing and Research with Human Subjects**

    Question: For crowdsourcing experiments and research with human subjects, does the paper include the full text of instructions given to participants and screenshots, if applicable, as well as details about compensation (if any)?

    Answer: [NA]

    Justification: As mentioned in previous questions this is not applicable for this paper and all datasets evaluated are popular large-scale networks from social media that have been widely evaluated on in recent years in the network science community.

Guidelines:

- The answer NA means that the paper does not involve crowdsourcing nor research with human subjects.
- Including this information in the supplemental material is fine, but if the main contribution of the paper involves human subjects, then as much detail as possible should be included in the main paper.
- According to the NeurIPS Code of Ethics, workers involved in data collection, curation, or other labor should be paid at least the minimum wage in the country of the data collector.

15. **Institutional Review Board (IRB) Approvals or Equivalent for Research with Human Subjects**

Question: Does the paper describe potential risks incurred by study participants, whether such risks were disclosed to the subjects, and whether Institutional Review Board (IRB) approvals (or an equivalent approval/review based on the requirements of your country or institution) were obtained?

Answer: [NA]

Justification: This paper does not involve any research or testing of human subjects. All training and testing procedures are solely conducted on publically released benchmark datasets that do not involve intervention from human subjects.

Guidelines:

- The answer NA means that the paper does not involve crowdsourcing nor research with human subjects.
- Depending on the country in which research is conducted, IRB approval (or equivalent) may be required for any human subjects research. If you obtained IRB approval, you should clearly state this in the paper.
- We recognize that the procedures for this may vary significantly between institutions and locations, and we expect authors to adhere to the NeurIPS Code of Ethics and the guidelines for their institution.
- For initial submissions, do not include any information that would break anonymity (if applicable), such as the institution conducting the review.

