# OpenReview forum: "Non-Euclidean Mixture Model for Social Network Embedding"
_NeurIPS.cc/2024/Conference — NeurIPS 2024 poster_

### Official Review · Reviewer_BsLx · 2024-06-18

**Soundness:** 3
**Presentation:** 3
**Contribution:** 2
**Rating:** 7
**Confidence:** 4

**Summary:**

This paper proposes NMM-GNN, a non-Euclidean mixture model that captures both homophily and hierarchies in social networks for embedding.

**Strengths:**

1.The paper is well-structured, clearly written, and easy to follow.

2.In the experiments section, the author compared baselines from different categories on multiple tasks (classification and link prediction). The results show that the NMM-GNN proposed in the paper consistently achieves the best performance.

**Weaknesses:**

1. The related work section does not cover most existing social embedding works. For example, many works, in addition to RaRE, also consider both the similarities and the social impact of nodes.

2. There is no clear proof or at least empirical experiments to demonstrate that it is more reasonable to embed both node similarity and social impact in spherical space instead of Euclidean space.

**Questions:**

1. My main concern is that social network embedding algorithms considering social impact are not novel. For example, when performing edge prediction, one can directly assume the probability of the edge is proportional to the degree of both nodes. I hope the author could further explain why the proposed method is novel enough in terms of encoding both similarity and impact.

2. Despite the experimental results and ablation study in the main article showing the good performance of the proposed method, I want to know how the speed of the method compares with existing baselines.

**Limitations:**

The author discussed the potential limitations of the paper in detail in the appendix.

---

> ### Author Rebuttal · Authors · 2024-08-07
>
> Thanks a lot for your insightful comments and feedback. Please find our response below to your questions.
>
> Q1: The related work section does not cover most existing social embedding works. For example, many works, in addition to RaRE, also consider both the similarities and the social impact of nodes.
>
> A1: Please refer to Table 9 in our main paper of “Category and Description of Baseline models” where we discuss and empirically evaluate against all the latest state-of-the-art models for social network embedding models. We cover 15 baseline models (up until the year of 2024) in all categories of learning models including (1) structural embedding models, (2) GNN embedding models, (3) homophily-based embedding models, and (4) mixture models. Our model consistently outperforms all the models for both classification and link prediction on four comprehensive metrics of Jaccard Index (JI), Hamming Loss (HL), F1 Score (F1), and AUC. Please refer to Tables 10, 12 (Appendix), and 14 (Appendix).
>
> Q2: There is no clear proof or at least empirical experiments to demonstrate that it is more reasonable to embed both node similarity and social impact in the spherical space instead of Euclidean space.
>
> A2: We have conducted extensive ablation studies, as shown in Table 10, which could be used as empirical experiments to justify our approach. In these studies, we evaluated several variations of the NMM model, using different non-Euclidean geometric spaces for homophily and social influence. Specifically, we compared the performance of models where E, S, and H denote Euclidean, Spherical, and Hyperbolic spaces, respectively. Our results demonstrate that modeling homophily in spherical space and social rank in hyperbolic space significantly outperforms other configurations, including those using Euclidean space.
>
> Q3: My main concern is that social network embedding algorithms considering social impact are not novel. For example, when performing edge prediction, one can directly assume the probability of the edge is proportional to the degree of both nodes. I hope the author could further explain why the proposed method is novel enough in terms of encoding both similarity and impact.
>
> A3: We would like to clarify and highlight that the novelty of our work lies in representing the factors in network science that explain how links are generated in the social network: homophily and social influence. As we observe through resulting topological patterns formed by homophily (cycles) and social influence (hierarchy), we have motivation to utilize the non-Euclidean geometric spaces of spherical/hyperbolic to model the resulting topologies. Specifically, we are among the first to not only jointly model for homophily/social influence, but also do so with the novel integration of multiple non-Euclidean geometric spaces used together (with novel space unification architecture) through an efficient Non-Euclidean Mixture Model (NMM). This addresses the dual aspects of homophily and social influence which we model in personalized way for each link of the social network in a unified framework, which, to the best of our knowledge, has not been previously explored.
>
> Q4: Despite the experimental results and ablation study in the main article showing the good performance of the proposed method, I want to know how the speed of the method compares with existing baselines.
>
> A4: In the main paper, we will be sure to include formal quantitative comparisons for computational efficiency. Regarding our model, NMM is highly efficient, with time complexity O(n * d), where n is number of nodes and d is dimension size. In comparison, here are the time complexity analyses for the remaining baseline models, mixture model of RaRE is O(n * d) which is comparable to the NMM mixture model, the GNN embedding models of GCN is O(n^2 * d + n * d^2), and GAT is O(n * d^2), the non-Euclidean GNN embedding models of k-GCN is O(k * [n^2 * d + n * d^2]) and HGCN is O(2d^2 + a * n * d^2) = O(a * n * d^2) where ‘a’ is the filter length, and NMM-GNN is O(n^2 * d + n * d^2) which is comparable to GNN embedding models. Moreover, we would like to point out that GraphVAE (of NMM-GNN) training is designed to be highly parallelizable, which allows for scalability. Moreover, our model is capable of learning on real-world, highly large-scale graphs on the order of millions of nodes and billion of edges, e.g., Friendster, while achieving the SOTA performance, which attests to its practical value to the network science community.

---

> > ### Comment · Reviewer_BsLx · 2024-08-09
> >
> > Thanks a lot for your response. I appreciate and agree with your response, and I will change my rating from 6 to 7.

---

> > > ### Author Response · Authors · 2024-08-11
> > >
> > > We sincerely appreciate your positive feedback and the increase in the score. We are glad that we were able to address your questions.

---

### Official Review · Reviewer_BmsX · 2024-07-04

**Soundness:** 4
**Presentation:** 4
**Contribution:** 3
**Rating:** 5
**Confidence:** 5

**Summary:**

The authors understand why links are generated through node-embedded representations of social networks. Specifically, spherical space is utilized to represent the homogeneity of nodes and hyperbolic space is utilized to represent the hierarchy and influence of nodes. By mixing these two spaces together, the corresponding link representation is finally obtained.

**Strengths:**

1.The writing is clear and easy to understand.
2.The experiments were adequate and demonstrated the power of the models

**Weaknesses:**

1.The idea of the article is not novel. Both hyperbolic and spherical spaces are geometric models that are very commonly used in the field of social networking.[1]
2.Although the authors are trying to explain linking relationships in terms of homogenization and hierarchy, however the learning of the model embedding is still unknown, which does not explain the emergence of social networks.
3.There has been much better work on embedding methods in hybrid spaces. [2]The authors don't contribute as much to this as their article suggests.
[1]Network geometry. Nature physics
[2]Motif-aware Riemannian Graph Neural Network with Generative-Contrastive Learning AAAI 2024
[3]Product Manifold Learning. AISTATS 2021

**Questions:**

1.How to analyze the reasonable causes of social network link generation from complex feature representations?
2.Can this model explain small-world networks in complex networks?

**Limitations:**

1.The model does not really give an adequate explanation of links in complex networks, but only from the perspectives of homogeneity and hierarchy.

---

> ### Author Rebuttal · Authors · 2024-08-07
>
> Q1: Idea is not novel. Hyperbolic/spherical spaces are are commonly used in social networking. [1] Network geometry. Nature physics
>
> A1: We would like to clarify and highlight that the novelty of our work lies in representing the factors in network science that explain how links are generated in the social network: homophily and social influence. We are among the first works to jointly model both factors in the network when comprehensively surveying the latest literature of work of 15 baseline models (up until the year of 2024) in all categories of learning models including (1) structural embedding models, (2) GNN embedding models, (3) homophily-based embedding models, and (4) mixture models. As we observe through resulting topological patterns formed by homophily (cycles) and social influence (hierarchy), we have motivation to utilize the non-Euclidean geometric spaces of spherical/hyperbolic to model the resulting topologies. Specifically, we are among the first to not only jointly model jointly for homophily/social influence, but also doing so with the novel integration of multiple non-Euclidean geometric spaces used together (with novel space unification architecture) through an efficient Non-Euclidean Mixture Model (NMM).
>
> Q2: Although authors are trying to explain linking relationships in terms of homogenization and hierarchy, however learning of the model embedding is still unknown, which does not explain the emergence of social networks.
>
> A2: We utilize feature learning through latent embeddings to represent nodes in terms of homophily and social influence factors. Homophily is based on feature similarity which can be captured through cosine similarity of embeddings, and social rank can be captured via norm space. Furthermore, these learned embeddings are highly interpretable. Specifically, in our visualizations of the network embeddings, we see that celebrity nodes are embedded towards the center of the Poincare disk, while nodes with lower social rank are embedded towards the boundary. Nodes of high homophily can be embedded closer to each other on the spherical space compared to what the hyperbolic or Euclidean spaces even allows for (better capturing cyclic influence). We will add this interpretable visualization to the paper.
>
> Q3: There has been much better work on embedding methods in hybrid spaces. [2] Motif-aware Riemannian Graph Neural Network with Generative-Contrastive Learning AAAI 2024. The authors don’t contribute as much to this as their article suggests.
>
> A3: The novelty of our work lies in representing the factors in network science that explain how links are generated in the social network: homophily and social influence which form resulting topological patterns formed by homophily (cycles) and social influence (hierarchy) motivating our use of non-Euclidean geometric spaces of spherical/hyperbolic. Specifically, we are among the first to not only jointly model jointly for homophily/social influence, but also doing so with the novel integration of multiple non-Euclidean geometric spaces used together (with novel space unification architecture) through an efficient Non-Euclidean Mixture Model (NMM).
>
> The paper you reference is not modeling social networks (critical for our problem of investigation) but rather some general purpose graphs Cora etc. (1) this model does capture laws of how links are formed in social networks (homophily/social influence), and (2) do not exhibit the types of resulting topologies from those factors. Also distinct/individual non-Euclidean geometric spaces are considered but rather does not explore how to jointly integrate different curvature non-Euclidean geometric spaces (to have joint influence from social network factors).
>
> Q4: How to analyze the reasonable causes of social network link generation from complex feature representations?
>
> A4: Please refer to sections 3.2.1 “Link prediction using homophily based distribution” and 3.2.2 “Link prediction using social influence based distribution” where we explain the probability distributions for our mixture model NMM by considering the impact of high/low homophily (+ noise/sparsity factor) in addition to high/low social influence (+ noise/sparsity factor). To empirically validate our claims and quality of our NMM mixture model’s learned embeddings, in our experiment results on Table 10, we rigorously evaluate the results of social network classification and link prediction against 15 other SOTA baseline models.
>
> Q5: Can model explain small-world networks in complex networks?
>
> A5: Definitely, which is substantiated through rigorous empirical evidence. We evaluate on numerous datasets including: social network datasets (domain focus of this research), citation networks (Appendix), and attributed networks (Appendix). For example, the social network datasets include BlogCatalog, LiveJournal, Friendster which are a balanced mix of small-scale and the largest scale size (node counts: of 10k, 5M, 66M respectively) and both directed and undirected. These numerous datasets show the generalizability performance and scalability of our approach as they are in all sizes for vertices (4K to 66M), edges, edge types, attributes, classes (6 to 500) etc. Please see Tables 8, 11, 13 of main paper for details.
>
> Q6: The model doesn't really give an adequate explanation of links in complex networks, but only from perspectives of homogeneity and hierarchy.
>
> A6: It is largely agreed that social network links are formed due to either homophily or social influence from the network science community through numerous years of research development. In fact, most existing embedding models are designed based on the homophily aspect of social networks [4, 5]. However, research of RaRE [6] and work of [7] show homophily is insufficient, and social influence is also critical in forming connections. This is due to popular nodes having direct influence in forming links [8]. Refer to our paper for [4],[5],[6],[7].

---

> > ### Comment · Reviewer_BmsX · 2024-08-08
> > **Response to Rebottal**
> >
> > Thanks for your response! I have known the contribution and novelty of your work. At last, I have raised my score.

---

> > > ### Author Response · Authors · 2024-08-11
> > >
> > > We sincerely appreciate your positive feedback and the increase in the score. We are glad that the discussion on the contribution and novelty of our work helped to address your questions.

---

### Official Review · Reviewer_3fkq · 2024-07-06

**Soundness:** 3
**Presentation:** 3
**Contribution:** 3
**Rating:** 6
**Confidence:** 3

**Summary:**

This paper proposes a new Graph-based non-Euclidean mixture model for social networks.
Under the assumptions that social network links are formed due to either homophily or social influence,
the homophily factor is modeled in spherical space and the social influence factor is in hyperbolic space.
The homophily regulated nodes lie on the surface of the spherical ball and the social influence-regulated
nodes lie on the open Poincare ball.
The projection to align these two spaces is also proposed.
The non-Euclidean GraphVAE is also integrated into the model.
Experiments show the proposed model outperforms other baselines.

**Strengths:**

- The proposed framework models both the homophily and social influence factors for social network generation.
- It is integrated into the non-Euclidean graph-based VAE to further improve performance.
- Experiments show the proposed model outperforms other SOTA baselines.

**Weaknesses:**

- The motivation of the space unification is unclear (See the questions below).

**Questions:**

- Why is the space unification necessary? Each node has two coordinates, in the spherical and the hyperbolic space. The link between nodes i and j is generated either from the spherical or hyperbolic proximity (it is the "explaining away" situation as explained in the RaRE paper [6] ). The homophily and social influence relationships may be unrelated or may even contradict to each other (you may hate your boss). Therefore the two coordinates may not necessarily be aligned. It would be appreciated if the authors would elaborate on this.

- The $\kappa$-GCN proposed in [39] also deals with spherical and hyperbolic spaces. I would like to know the similarities and/or differences between the two approaches as the two coordinates in the proposed model can also be regarded as a coordinate in the product space. The authors do not discuss them. They just compare the experimental performance.

- What is $i$ in the definition of $\log_0^H$ and $\exp_o^H$ ?

**Limitations:**

As for the computational efficiencies, more formal and quantitative comparisons would be necessary in the main texts.

---

> ### Author Rebuttal · Authors · 2024-08-07
>
> Thanks a lot for your insightful comments and feedback. Please find our response below to your questions.
>
> Q1: The motivation of the space unification is unclear: Why is the space unification necessary? Each node has two coordinates, in the spherical and the hyperbolic space. The link between nodes i and j is generated either from the spherical or hyperbolic proximity (it is the “explaining way” situation as explained in the RaRE paper [6]). The homophily and social influence relationships may be unrelated or may even contradict each other (you may hate your boss). Therefore, the two coordinates may not necessarily be aligned. It would be appreciated if the authors would elaborate on this.
>
> A1: We would like to clarify and highlight that the link between nodes i and j is a mixture model because each link is a weighted combination of influence from both spherical and hyperbolic spaces (not one or the other) as evidenced in Equation 6 of the paper. Hence, as shown in Figure 1b, the same node has 2 representations – one in the spherical space and one in the hyperbolic space, and because they represent the same underlying node, they need to be aligned. In the case of “you may hate your boss” if both you and your boss are not highly popular nodes e.g., celebrity nodes, then likely the social rank may be similar due to low social influence. At the same time, regardless of liking each other, you and your boss may share many similar characteristics like working at the same company, studied the same field like “computer science”, live in the same location/country, working on the same project problems etc. the homophily is still high by its very definition. Therefore, these two network factors do not contradict each other, but rather work together to explain how links are formed between users. We address this as well as scenarios of noise such as where nodes may still not be connected to each other even after exhibiting high homophily, as well as in the case they are not connected when having low social influence (or similar social rank). Please refer to sections 3.2.1 “Link prediction using homophily based distribution” and 3.2.2 “Link prediction using social influence based distribution” where we explain this and how our distribution models also represent factors to control sparsity of the network.
>
> Q2: The K-GCN proposed in [39] also deals with spherical and hyperbolic spaces. I would like to know the similarities and/or differences between the two approaches as the two coordinates in the proposed model can also be regarded as a coordinate in the product space. The authors do not discuss them. They just compare the experimental performance.
>
> A2: First, our model is not in the product space (e.g., where the entire model belongs to a cartesian product of non-Euclidean geometric spaces by default). Rather, our work is in a category called mixed space model that uses a multi-geometric space framework where different portions of the graph may possibly belong to different spaces (based on the amount of impact each of homophily and social influence has for that personalized pair of nodes). In the extreme case (Case 1) where only social influence is at play (e.g., weight of homophily representation is learned close to 0), the hyperbolic space will be used. On the other hand if only homophily is at play (Case 2)  e.g., weight of homophily representation is learned close to 0, the spherical space will be used. In the normal case of both factors at play (Case 3), then both spaces will be used and can be jointly aligned with our space alignment mechanism. When using product space, Cases 1/2/3 will all not be distinguished from each other (though that would be a better representation) as all cases will be modeled by one complex non-Euclidean geometric space as a cartesian product of spherical and hyperbolic spaces.
>
> Q3: What is ‘i’ in the definition of log_0^H and exp_0^H?
>
> A3: The ‘i’ is the mathematical symbol - sqrt(-1) (the formal term: imaginary number). i (which is in italics) is notation e.g., z_i, which is referring to the i-th node.
>
> Q4: As for the computational efficiencies, more formal quantitative comparisons would be necessary in the main texts.
>
> A4: Thank you for your comment. In the main paper, we will be sure to include formal quantitative comparisons for computational efficiency. Regarding our model, NMM is highly efficient, with time complexity O(n * d), where n is number of nodes and d is dimension size. In comparison, here are the time complexity analyses for the remaining baseline models, mixture model of RaRE is O(n * d) which is comparable to the NMM mixture model, the GNN embedding models of GCN is O(n^2 * d + n * d^2), and GAT is O(n * d^2), the non-Euclidean GNN embedding models of k-GCN is O(k * [n^2 * d + n * d^2]) and HGCN is O(2d^2 + a * n * d^2) = O(a * n * d^2) where ‘a’ is the filter length, and NMM-GNN is O(n^2 * d + n * d^2) which is comparable to GNN embedding models. Moreover, we would like to point out that GraphVAE (of NMM-GNN) training is designed to be highly parallelizable, which allows for scalability. Moreover, our model is capable of learning on real-world, highly large-scale graphs on the order of millions of nodes and billion of edges, e.g., Friendster, while achieving the SOTA performance, which attests to its practical value to the network science community.

---

> > ### Comment · Reviewer_3fkq · 2024-08-10
> >
> > Thanks for the detailed explanation.
> >
> > As for Q1,  I am aware that the proposed model is a mixture model.
> >
> > What I wanted to know is the reason why the space unification is necessary (the motivation of the space unification loss). The space unification regularization term ensures that two geometric spaces are aligned together to make sure the two embeddings of the same node correspond to each other. Why do the two embeddings of the same node have to be close to each other?
> >
> > If $\boldsymbol{Z}_i^H \approx \boldsymbol{Z}_i^S$ and  $\boldsymbol{Z}_j^H \approx \boldsymbol{Z}_j^S$,
> >
> > I wonder $\mathrm{dist}_h(\boldsymbol{z}_i^H,\boldsymbol{z}_j^H)
> > \approx$ $\mathrm{dist}_s(\boldsymbol{z}_i^S,\boldsymbol{z}_j^S)$
> > (I know this is oversimplified, but you may still say that the former distance is small then the latter distance is also small)
> >
> > then I wonder
> >
> > $p(e_{ij}=1\vert \mathrm{dist}_h(\boldsymbol{z}_i^H,\boldsymbol{z}_j^H))$
> >
> > may tend to be equivalent to
> >
> > $p(e_{ij}=1\vert \mathrm{dist}_s(\boldsymbol{z}_i^S,\boldsymbol{z}_j^S))$

---

> > > ### Author Response · Authors · 2024-08-10
> > > **Response to Follow-Up Question from Reviewer 3fkq**
> > >
> > > Thanks for your follow-up question. Please find our response below:
> > >
> > > We would like to address a misunderstanding in your interpretation of the paper: It seems that you thought the alignment is to enforce the embeddings the same in two spaces, which is untrue. We require the projection of Z_i^H is close to i's embedding in spherical space Z_i^S. Note, it's impossible to equate these two directly as they are in different geometric spaces.
> > >
> > > In this case, the two distances (of spherical and hyperbolic spaces) are also different from each other. Note in our paper dist_h(zHi , zHj ) = |norm(zHi ) − norm(zHj )| is not geodesic distance in hyperbolic metric space. It is defined based on their norm to reflect their rank difference.
> > >
> > > Without the alignment, Z_i^H has too much degree of freedom, which can move freely as long their norm kept the same.
> > >
> > > We also have an ablation study on this part. Please refer to Appendix section's Figure 2 (c), where we show the quality of using the space unification component (both with and without), and we observe that the performance improves consistently on across all datasets with the space unification component.

---

> > > > ### Comment · Reviewer_3fkq · 2024-08-11
> > > >
> > > > Thanks for the clarification.
> > > > The geodesic relationship between zHi and zHj in the hyperbolic metric space does not directly govern the link probability, but only the norm difference matters.
> > > > I will change my rating from 4 to 6.

---

> > > > > ### Author Response · Authors · 2024-08-11
> > > > >
> > > > > We sincerely appreciate your positive feedback and the increase in the score. We are glad that the additional clarifications we provided have helped to address some misunderstandings of the paper and demonstrate the importance of the space alignment. We will rewrite some of our text to include these insights in the next revision, in case other readers also also have this confusion.

---

### Official Review · Reviewer_iEGq · 2024-07-12

**Soundness:** 3
**Presentation:** 3
**Contribution:** 3
**Rating:** 7
**Confidence:** 3

**Summary:**

This work addresses the embedding of social networks with downstream tasks such as link prediction in mind. In this manuscript the authors propose to model the link as the mixture of two factors of node embedding, Spherical and Hyperbolic. Concretely, two kinds of node embedding are unified into a single loss by projecting from hyperbolic space into spherical space, and jointly trained in the framework of VAE on Graaph. The authors also conduct experiments against a wide range of baseline methods to highlight the capacity of the proposed method.

**Strengths:**

The idea is novel. Overall things are good. The idea of bridging both is novel, and according to experiments performance better than purely hyperbolic embedding. Writing is overall clear and, although there are many details, it’s generally feasible to follow. Extensive comparison with a wide range of baselines from different categories. Good discussion of limitations.

**Weaknesses:**

The major issue is on limited datasets in experiments. I would suggest the authors consider other datasets used in previous works, such as synthetic datasets, citation networks (PubMed, wikipedia citation,  DBLP, Microsoft Academic Graph ) and other social networks (e.g. Twitter).

Also another factor limiting the impact of the proposed method is that the proposed method does not work on the whole-graph level, and thus only applies to node classification and link prediction tasks. If by any extension the embeddings could be aggregated to the whole-graph leve, there could be more significance (e.g. tasks like molecular classification, protein-to-protein interactions)

**Questions:**

N/A

**Limitations:**

Well addressed.

---

> ### Author Rebuttal · Authors · 2024-08-07
>
> Thanks a lot for your insightful comments and feedback. Please find our response below to your questions.
>
> Q1: The major issue is on limited datasets in experiments. I would suggest the authors consider other datasets used in previous works, such as synthetic datasets, citation networks (PubMed, wikipedia citation, DBLP, Microsoft Academic Graph) and other social networks (e.g., Twitter)
>
> A1: We would like to clarify that the primary goal of our work is to explain how links are generated in social networks e.g., user to other users. As such, for datasets, in the main paper we specifically showcase results on social network datasets as that is the domain focus of this research. However, in the Appendix (Tables 12 and 14), we also include experiments for the citation networks of Wikipedia datasets, to show that our model can also benefit other networks, as well as attributed network datasets.  In total, we evaluate on seven datasets. The social network datasets include BlogCatalog, LiveJournal, Friendster which are the most prominent and recent large-scale networks from the latest research in the literature, and which are a balanced mix of small-scale and the largest scale size (node counts: of 10k, 5M, 66M respectively) and both directed and undirected. For citation networks, we evaluate on Wikipedia Clickstream and Wikipedia Hyperlink. For attributed networks, we evaluate Facebook and Google+ (containing ~16K attributes), and these numerous datasets show the generalizability performance and scalability of our approach.
>
> Q2: Also another factor limiting the impact of the proposed method is that the proposed method does not work on the whole-graph level, and thus only applies to node classification and link prediction tasks. If by any extension the embeddings could be aggregated to the whole-graph level, there could be more significance (e.g., tasks like molecular classification, protein-protein interactions)
>
> A2:
> Our work can definitely be generalized to the graph-level because our method learns to represent the social science network factors based on topologies in the graph on clusters of nodes and edges. Thus, if the cluster of nodes and edges comprised of the entire graph and we subsequently applied graph pooling per node embedding (that we currently learn), we can reduce the embedding from node level to graph level. In this way, homophily/social influence can be modeled at the graph level. That said, it is unclear whether graph-level modeling would be specifically useful/interpretable for social network embedding models (as compared to node-level modeling). This is due to the social network setting requiring links to be generated that are per node (not at the graph level), since a user is recommended to another specific user in the practical social network setting. This is different from other network domains like molecular classification where the entire graph represents one molecule e.g., atoms form individual nodes and chemical bonds form the edges. On the other hand, one node represents one user (unlike molecular graphs). We’d also like to clarify that for this reason, node-level learning is in fact consistent with the recent state-of-the-art NN methods in the network science community though our model can still be generalized to learning at the graph-level.

---

### Author Rebuttal · Authors · 2024-08-07

Dear Reviewers: Thank you for your time in reading our paper and for your useful comments/questions/suggestions on our paper. We have responded individually to each reviewer, however, we are also including a general summary answer to some common questions:

Datasets: In the main paper we specifically showcase results on social network datasets as that is the domain focus of this research. However, in the Appendix (Tables 12 and 14), we also include experiments for the citation networks of Wikipedia datasets, to show that our model can also benefit other networks, as well as attributed network datasets.  In total, we evaluate on seven datasets ranging from small-scale to large scale for nodes, edges, types of edges, attributes, classes etc. on a very diverse range of datasets. These numerous datasets show the generalizability performance and scalability of our approach (which we rigorously evaluate on 15 SOTA baseline models of four different learning method categories on both social network classification and link prediction on 4 metrics).

Computational complexity: In the main paper, we will be sure to include formal quantitative comparisons for computational efficiency. Regarding our model, NMM is highly efficient, with time complexity O(n * d), where n is number of nodes and d is dimension size. In comparison, here are the time complexity analyses for the remaining baseline models, mixture model of RaRE is O(n * d) which is comparable to the NMM mixture model, the GNN embedding models of GCN is O(n^2 * d + n * d^2), and GAT is O(n * d^2), the non-Euclidean GNN embedding models of k-GCN is O(k * [n^2 * d + n * d^2]) and HGCN is O(2d^2 + a * n * d^2) = O(a * n * d^2) where ‘a’ is the filter length, and NMM-GNN is O(n^2d + nd^2) which is comparable to GNN embedding models. Moreover, we would like to point out that GraphVAE (of NMM-GNN) training is designed to be highly parallelizable, which allows for scalability. Moreover, our model is capable of learning on real-world, highly large-scale graphs on the order of millions of nodes and billion of edges, e.g., Friendster, while achieving the SOTA performance, which attests to its practical value to the network science community.

Novelty: We would like to clarify and highlight that the novelty of our work lies in representing the factors in network science that explain how links are generated in the social network: homophily and social influence. We are in fact one of the first works to jointly model both factors in the network when comprehensively surveying the latest literature of work of 15 baseline models (up until the year of 2024) in all categories of learning models including (1) structural embedding models, (2) GNN embedding models, (3) homophily-based embedding models, and (4) mixture models. As we observe through resulting topological patterns formed by homophily (cycles) and social influence (hierarchy), we have motivation to utilize the non-Euclidean geometric spaces of spherical/hyperbolic to model the resulting topologies. Specifically, we are among the first to not only jointly model jointly for homophily/social influence, but also doing so with the novel integration of multiple non-Euclidean geometric spaces used together (with novel space unification architecture) through an efficient Non-Euclidean Mixture Model (NMM). This addresses the dual aspects of homophily and social influence which we model in a personalized way for each link of the social network in a unified framework, which, to the best of our knowledge, has not been previously explored.

Analyzing Causes of Social Network Link Generation: As mentioned in detail in the paper, it is widely agreed from the network science community that two factors (homophily and social influence) affect how links are generated in the social network. Please refer to sections 3.2.1 “Link prediction using homophily based distribution” and 3.2.2 “Link prediction using social influence based distribution” where we explain the probability distributions for our mixture model NMM by considering the impact of high/low homophily (+ noise/sparsity factor) in addition to high/low social influence (+ noise/sparsity factor). To empirically validate our claims and quality of our NMM mixture model’s learned embeddings, in our experiment results on Table 10, we evaluate the results of social network classification and link prediction for Jaccard Index (JI), Hamming Loss (HL), F1 Score (F1), and AUC. Our model is comprehensively evaluated against 15 other state-of-the-art baseline models belonging to four different categories of learning representations.

---

### Decision · Program_Chairs · 2024-09-25

**Decision:**

Accept (poster)

**Comment:**

The paper proposes a graph embedding methodology in which link generation is modeled via a mixture model that leverages non-Euclidean geometric spaces. The proposed approach has merits that the reviewers have highlighted. These mostly include novel aspects of the methodology, such as modeling both homophily and social influence, as well as well-executed experiments.  The response of the authors to the different questions raised further clarified several aspects of the paper. Therefore, I recommend accepting the paper.